# Observations of cross-shelf transport due to internal wave pumping on the Bay of Biscay shelf

Adèle Moncuquet[1], Nicole L. Jones[2], Lucie Bordois[3], François Dufois[1], and Pascal Lazure[4]

[1]Institut Français de Recherche pour l'Exploitation de la Mer, IFREMER/DYNECO/DHYSED, Brest, France
[2]School of Earth and Oceans and Oceans Institute, University of Western Australia, Perth, WA, Australia
[3]Service Hydrographie et Océanographie de la Marine, Brest, France
[4]Institut Français de Recherche pour l'Exploitation de la Mer, LOPS (UMR6523 CNRS/IFREMER/IRD/UBO), Brest, France

**Correspondence:** Adèle Moncuquet (adele.moncuquet@gmail.com)

**Abstract.** Coastal cross-shelf transport drives the redistribution of sediment, nutrients and pollutants on continental shelves. Here, the cross-shelf volume flux is quantified from in situ measurements at two coastal sites on the Bay of Biscay (BoB) shelf. At both sites, a semidiurnal internal tide propagates onshore, and mode 1 nonlinear internal wave packets are observed. The Eulerian and Stokes drift contributions to the subtidal cross-shelf transport are estimated along density layers from ADCP, temperature sensors, and CTD measurements at 62 m water depth on the Landes plateau or SE-BoB (44°N) and 47 m water depth on the Armorican shelf or N-BoB (47°N). The Stokes drift transport has possible contributions from the internal tides, nonlinear internal waves and the surface waves and tide. At both sites, the vertical profile of the month-long averaged Stokes drift volume flux matches the shape of the semi-analytical Stokes drift volume flux due to a linear mode-1 internal tide. We demonstrate that nonlinear internal wave events can also contribute to the Stokes drift volume flux. We thereby attribute the Stokes drift volume flux at the study sites to internal wave pumping (IWP). At both sites, the IWP is responsible for a near-seabed onshore volume flux during stratified conditions and spring internal tides, that is equivalent to a wind-driven upwelling event generated by a 4 m/s wind. At N-BoB (47°N), IWP is the main contributor to the total cross-shelf volume flux under stratified conditions and a spring internal tide. At SE-BoB (44°N), IWP augments the near-seabed onshore volume flux during upwelling events and maintains a near-seabed onshore volume flux even during downwelling events.

## 1 Introduction

Transport across continental shelves is a key driver of nutrient, sediment, pollutant and physical properties of water masses. Offshore transport from the shelf to the sea can also impact the global circulation and represents one of the $CO_2$ pumps from the atmosphere towards the deep ocean (Huthnance et al., 2009). Cross-shelf transport may be driven by processes of various temporal and spatial scales such as eddies, fronts, Ekman transport, tidal pumping, surface waves, internal tides and their Stokes drift and nonlinear internal waves (Huthnance, 1995; Bowden, 1983). The contribution of internal wave cross-shelf transport remains poorly quantified due to the challenge of observing this process.

Observations of both nonlinear internal waves and internal tides have demonstrated a net volume transport from the open ocean to the shelf seas (Inall et al., 2001; Zhang et al., 2015). In theory, steady linear internal waves are not expected to

generate net transport because their Eulerian and Stokes transport cancel in an inviscid ocean under rotation or in the presence of a sloping seabed connected to a land boundary (Spingys et al., 2020). However, the two flows may not exactly cancel and transport can occur for unsteady waves (when there is a temporal evolution of the currents) or when diapycnal mixing occurs (Spingys et al., 2020), which is common for internal waves propagating upslope (Henderson, 2016). The theoretical Stokes drift vertical profile consists of three layers. It can be divided into a bolus transport, in the wave direction in the surface and bottom layers and an opposite shear transport in the intermediate layer (Henderson, 2016; Spingys et al., 2020). A net transport due to internal tide Stokes drift has been computed from observations at three different shelves (New Zealand, European Malin and Celtic shelves) (Spingys et al., 2020). Close to the Bay of Biscay, in the Celtic Sea, internal tide Stokes transport by internal waves on the Celtic shelf was estimated at 0.33 $m^2.s^{-1}$ in the bottom layer, -0.43 $m^2.s^{-1}$ in the interior, and 0.15 $m^2.s^{-1}$ in the surface layer (Huthnance et al., 2022).

Nonlinear internal waves range from steepened internal tides to packets of high frequency waves. Alongshore observations on Dongsha Atoll have demonstrated the upslope transport of cold water by a combination of tidal bores and nonlinear internal waves (Davis et al., 2020). Observations at Monterey Bay (Californian shelf) demonstrate the transport of cold, hypoxic deep water by tidal bores that is modulated by wind-driven upwelling events (Walter et al., 2014). Evidence of internal tide bore transport of planktonic larvae at the Southern Californian Bight were observed by Pineda (1991). NLIWs of elevation, or boluses, trap water masses within their core and transport these masses appreciable distances (Gil and Fringer, 2016). Net transport due to NLIWs of depression has been quantified on the Washington shelf, where regular NLIWs of different types generate onshore mass transport equivalent to Ekman transport (1 $m^2.s^{-1}$) (Zhang et al., 2015); the New Jersey Shelf , where the average daily NLIWs onshore transport was estimated as (0.3 $m^2.s^{-1}$) (Shroyer et al., 2010) and the Malin Shelf, where the summertime sustained transport was estimated as 0.3 $m^2.s^{-1}$ (Inall et al., 2001).

In the Bay of Biscay (BoB), mass transport over the shelf has previously been attributed to along-shore regional circulation, wind forcing and thermohaline circulation induced by strong river inputs (mainly the Loire, Gironde and Adour rivers). Over the shelf, the barotropic tidal currents increase from south (10 cm.s$^{-1}$ for M2 tidal current) to north (70 cm.s$^{-1}$ at 48°N) (Le Cann, 1990). Surface waves induce transport through Stokes drift, but their effect remains constrained to near the surface and quickly decreases with depth, with a reduction to 1/23 of the wave motion at a depth of half a wavelength below the surface in deep water (Bowden, 1983). Long-term (subtidal) transport due to internal waves has not previously been assessed at the Bay of Biscay shelf. Transport in the BoB is mainly wind driven and seasonally fluctuates (Le Boyer et al., 2013). On the south-eastern part of the BoB, cross-shelf transport is dominated by wind and bottom Ekman transport (Huthnance et al., 2009). Internal waves are expected to be numerous and highly energetic in summer on the Armorican shelf as the BoB shelf break is a hot spot for internal waves (Baines, 1982; Pingree and New, 1991; Gerkema, 2001; Xie et al., 2015; Barbot et al., 2021). The southern shelf of the BoB presents the smallest barotropic tide of the shelf (Koutsikopoulos and Cann, 1996). However, currents associated with internal waves up to 3 times the barotropic current were observed during summer on the southern shelf of the BoB (Moncuquet et al., 2025). NLIW packets were a common feature with a variety of wave responses depending on the vertical stratification. On the Bay of Biscay shelf the transport induced by internal waves is currently unknown.

In this paper, we quantify the Stokes drift cross-shelf subtidal volume flux using isopycnal coordinates at the two Bay of Biscay locations during the summer period. Here we define the Stokes drift transport as the difference between the total subtidal volume transport and the subtidal Eulerian transport; it can include contributions from the surface tide, surface waves, internal tide and nonlinear internal waves. We first describe the measurements at each site and the method used to compute both the Stokes drift and Eulerian subtidal volume transport. In the results section, we describe the observed dynamics at each site. We then present the monthly-averaged transport at each site. We compare the results at each site with the semi-analytical formulation for Stokes transport by a linear M2 internal tide (Franks et al., 2020). In the final section, we describe the time evolution of the fluxes at each site and link it to the observed dynamics.

## 2  Methods

The two observation sites were on the Armorican shelf over the northern part of the BoB (N-BoB) and on the Aquitaine shelf over the southeastern part of the Bay (SE-BoB) (Figure 1). The data sets are from two distinct campaigns (2017 and 2022) that were executed during northern hemisphere summer periods. We applied identical cross-shelf transport analysis at both sites.

### 2.1  Mooring sites

The Bay of Biscay is an open bay with a steep shelf break, located on the western part of the European margin. The BoB extends from the Celtic Sea to the Iberian Peninsula. The BoB shelf can be divided into three shelves of decreasing width towards the south (Figure 1). The Armorican shelf is between 100-150 km wide and is located in the north. The Aquitaine shelf is between 46°N and the Capbreton canyon (Figure 1a). The narrow Spanish shelf is perpendicular to the Aquitaine shelf and spreads along the Spanish coast (Figure 1a). Along the shelves, currents and stratification are forced by different mechanisms (Koutsikopoulos and Cann, 1996). The surface tide is mainly semi-diurnal and its intensity decreases towards the south (Le Cann, 1990). The two main freshwater inputs are from the Loire and the Gironde rivers and are reduced during the summer months (Puillat et al., 2004). The wind is the main driver of the currents over the shelf (Le Boyer et al., 2013) and on average the regional circulation speed is less than 2.5 $cm.s^{-1}$ during summer and increases in winter.

### 2.1.1  South-eastern BoB site

The SE-BoB has a double shelf break and is defined by the Cap Ferret Canyon in the north and the Capbreton Canyon in the south (Figure 1a). At this location, barotropic tides are small compared to the rest of the Bay and oriented in the cross-shelf direction (Figure 1a). The summer stratification is strong in the region, with surface temperatures that can reach 26°C, whereas bottom temperatures remain around 12°C along the 80 m isobath (Koutsikopoulos and Cann, 1996). The stratification may temporally be affected by upwelling along the Landes coast (Valencia et al., 2004; Froidefond et al., 1996) and by downwelling formed along the Spanish coast (Batifoulier et al., 2012). Due to the coast shape, westerly winds along the Spanish coast induce a pressure gradient along the French coast, generating poleward jets and ultimately increasing the bottom temperature around the 54 m isobath along the Landes shelf (Batifoulier et al., 2012). The cross-shelf circulation in summer is usually sheared in

the vertical and the current inversion corresponds to the pycnocline position (Le Boyer et al., 2013). The Cap Ferret Canyon has been identified as a region with strong internal wave generation (Barbot, 2018). Along 44°N, the internal tide and nonlinear internal waves propagate in the cross-shore direction and generate currents more than 3 times the barotropic tidal current (Moncuquet et al., 2025).

Here we used a near-coastal mooring that observed temperature and velocity in 62 m water depth, hereafter referred to as SE-BoB and 2C in (Moncuquet et al., 2025). The ETOILE campaign was initially designed to study high-frequency internal waves along the SE-BoB shelf (Lazure and Puillat, 2017). Here, we used the co-located measurements of temperature and velocity gathered during 07/09/2017 and 07/30/2017 (2C mooring - 44°00 N / 1°31 W). The mooring line had 6 temperature sensors (ADT7320) and 6 pressure sensors (MS583730BA01-50) with an acquisition frequency of 1 min. The vertical positions of the sensors were regularly spaced every 10 m, between 0.7 meters above the bottom (mab) and 47.5 mab (Figure 1d.e). The accuracy of the temperature sensors was 0.1 °C, the uncertainty on pressure sensors was 0.1 m (Lazure et al., 2019). Conductivity and temperature vertical profiles were measured using a moving vessel profiler (MVP) between 25-29 July 2017 to compute the density-temperature relationship (Figure A1, A2). In a previous paper using the same dataset (Moncuquet et al., 2025), the vertical distribution of the sensors was shown to be satisfactory to capture the stratification. A comparison of the temperature profiles from the thermistors and the higher-resolution MVP measurements is presented in Figure 7 of that study. A bottom-mounted ADCP (RDI Sentinel 300kHz) was deployed with an acquisition frequency of 0.5 Hz and a vertical bin resolution of 1 m. The ADCP's single-ping horizontal standard deviation in this configuration is 0.08 m/s (Figure 9 in (Instruments, 2009)).

### 2.1.2 Northern BoB site

The second site was located north of Belle-Ile island (Figure 1) at 47.1°N, 4°W on the Armorican shelf. The presence of a homogeneous cold water mass ($\sim$ 12 - 13 °C) called a cold pool or "bourrelet froid" is one of the main hydrological features in the northern part of the BoB. This cold water mass, centred around 100 m water depth, extends along the northern coast between 48.5°N and around 46.5°N with low seasonal and inter-annual variability. Its formation depends mainly on the thermal exchange near the surface, the intensity of the barotropic tidal currents, and occasional freshwater inputs that allow strong stratification which isolates the bottom layer and reduces the downward heat diffusion (Puillat et al., 2004). The winds at N-BoB are more intense than at SE-BoB (Koutsikopoulos and Cann, 1996), resulting in the deepening of the surface mixed layer. The surface mixed layer can also be affected by the surface freshwater layer originating from the Loire and Gironde rivers. However, summer salinity anomalies are small, as flows reach their annual minima and plumes are stretched offshore and southwards under the influence of prevailing north-westerly winds (Lazure and Jegou, 1998).

The SOLIBOB campaign was designed to study NLIWs close to Belle-Ile (Lazure, 2022). The internal tide is generated along the shelf break, offshore of the mooring site (Barbot et al., 2021; Pairaud et al., 2010; Pichon et al., 2013; Xie et al., 2015). NLIWs have frequently been observed along the north Armorican slope (Pingree and Mardell, 1985). This site is also close to the mid-shelf mud belt called the "Grande Vasière". Here we use co-located measurements of temperature and velocity gathered between 08/27/2022 and 09/30/2022 at 47 m water depth (M1 mooring - 47°15.733'N / 03°09.377'W). The Mastodon

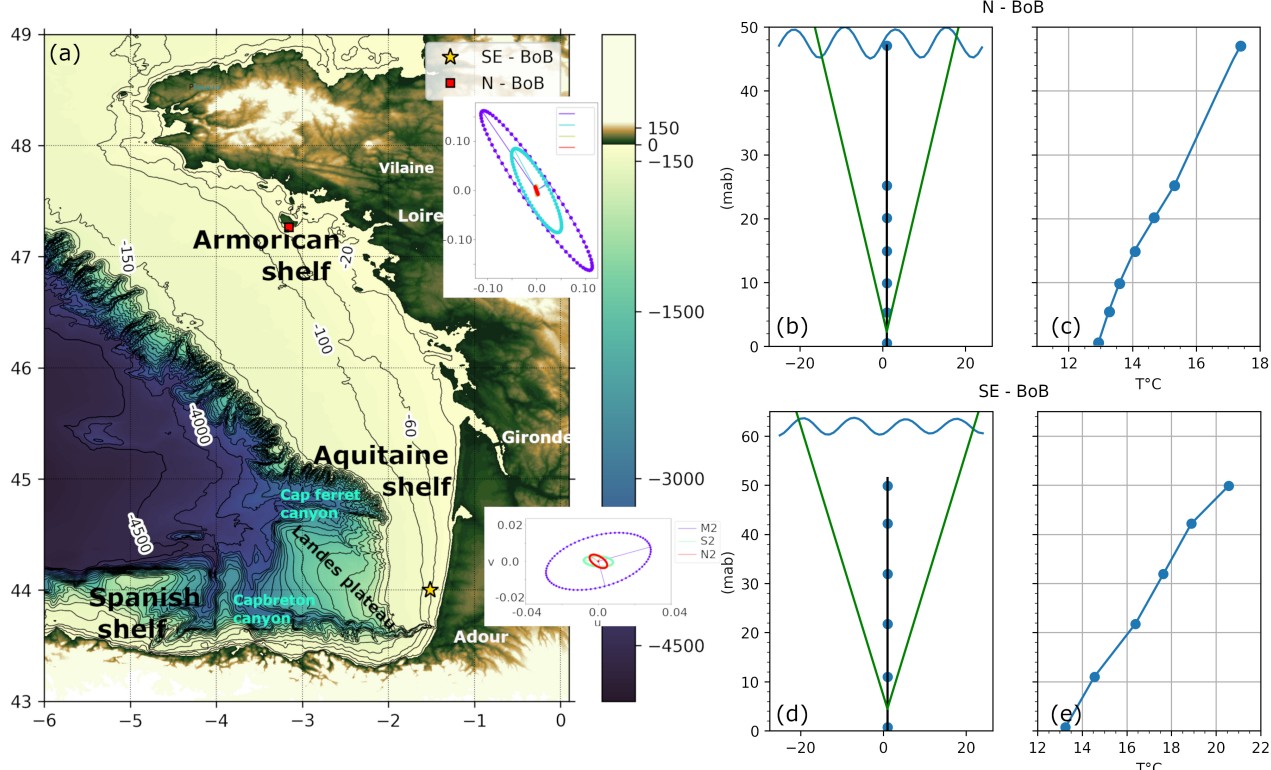

**Figure 1.** (a) BoB bathymetric map (colours) and mooring positions (the star corresponds to the SE - BoB experiment and the red square to the N - BoB experiment). The tidal ellipses are computed from the barotropic current at the mooring locations and show the contribution of the main tidal frequency at each site. Temperature profiles and mooring schematics are added for NE-BoB (b, c) and SE-BoB (d, e). The mooring schematics (b and d) show the spread of the ADCP beams in green solid lines and the position of the temperature and pressure sensors (blue solid circles). The height of each point is given in meters above the bottom (mab). The temperature profiles (c and e) show the summertime average temperature profiles from the moorings.

mooring line had 6 temperature (ADT7320) and pressure sensors (MS583730BA01-50) with an acquisition period of 2 min. The vertical position of the sensors was on average 5.3, 9.9, 14.9, 20.1, 25.2 and 47.1 m above the bottom (mab) (Figure 1b.c).

A bottom-mounted ADCP (Sentinel V50) was deployed with an acquisition frequency of 0.5 Hz and a vertical bin resolution of 0.5 m. In this configuration the single ping standard deviation is 0.07 $m.s^{-1}$. The temperature profiles were constructed from the 6 temperature sensors and the ADCP temperature sensor. The accuracy of the temperature sensors was 0.1 °C, the uncertainty on pressure sensors was 0.1 m (Lazure et al., 2019).

## 2.2 Data analysis for hydrographic conditions

### 2.2.1 Temperature and velocity fields

We linearly interpolated temperature and cross-shore velocity measurements onto a uniform vertical grid with 0.5 m spacing. Velocities were time-averaged on the same temporal resolution as the temperature sensors which varied between sites. The temporal resolution was 1 min for the SE-BoB mooring and 2 min for the N-BoB mooring. The coordinates of the current for the N-BoB site were rotated counter-clockwise by 60° to align with the shelf slope. Hence, the cross-shore velocity $u$ is positive when directed in the NE direction for this site. At SE-BoB, the eastward velocity corresponds to the cross-shore component; therefore, no rotation was applied.

We defined the background fields as the tidally filtered temperature and velocity fields. To filter the tidal components we used a Butterworth 24 h low pass filter of order 3. The filtered data are denoted $< . >$. From the background temperature field, we computed the stratification and the modal profile.

### 2.2.2 Stratification

To define the stratification over the whole water column we set a constant temperature above the highest sensor (i.e. no gradient in the surface layer). Salinity and temperature profile measurements were made in the vicinity of the moorings during the deployment or the recovery campaigns. At both locations, the relationship between density and temperature was linear (see Annex : Figure A1 and Figure A2). We hypothesized that the salinity was fairly constant over the whole month so that we could derive density from temperature measurements at each mooring line. To first order, this is reasonable as the influence of freshwater is weak, confined to the surface and is less important at this time of year for both sites (Puillat et al., 2006). Moreover, the main stratification profile uncertainty is expected to come from the low vertical resolution of the temperature measurements (10 m). We used equation (1) to compute the density field at SE-BoB and equation (2) at N-BoB.

$$\rho = -0.31T + 1031.19 \tag{1}$$

$$\rho = -0.256T + 1030.1 \tag{2}$$

The Brunt-Väisälä frequency $N$ was computed without the compressible effect, using the following equation :

$$N^2 = -\frac{g}{\rho}\left(\frac{d\rho}{dz}\right) \tag{3}$$

### 2.2.3 Modal structure

From $N^2$ we computed the time evolution of the first vertical mode $\phi_1$ and the propagation speed of the semi-diurnal internal tide $c_{M2}$ from the complete Sturm-Liouville equation (4), with the Coriolis coefficient $f = 10^{-4}.s^{-1}$ at SE-BoB (44°N) and

at N-BoB (47.2°N).

$$\frac{d^2\phi}{dz^2} + k^2 \frac{N^2 - \omega^2}{\omega^2 - f^2}\phi = 0 \tag{4}$$

We use subscript $\phi_1$ to denote the first mode.

### 2.2.4 Nonlinear internal wave identification

We identified the NLIWs propagating in the cross-shore direction from the temperature and velocity fields. We interpolated the vertical position of a set of isotherms with 1°C difference from temperature and pressure measurements. We high-pass filtered the vertical position of all the isotherms at 2 h to retain the high-frequency oscillations. We determined the time when the high-frequency oscillations of three isotherms reached an amplitude of $0.1\,H$, where $H$ is the water depth. This is a strong restriction and we acknowledge we will underestimate the number of NLIW.

An event is noted as a NLIW if the vertical velocity changes sign during the time the three isotherms reach the maximum amplitude and go back to their initial position. To define NLIW polarity we used the horizontal, cross-shore, baroclinic velocity and vertical velocity measurements as defined by Shroyer et al. (2009).

### 2.3 Quantification of cross-shelf volume flux using temperature coordinates

Volume fluxes were computed over a time-varying vertical section of the water column defined between two isotherms of 1
°C difference. We used the linearly interpolated temperature and velocity field between the ADCPs first cell (*i.e.,* 4.5 mab (SE-BoB) and 2.8 mab (N-BoB)) and 10 m below the surface. The vertical section of integration depends on how the isotherms are filtered in time. This method was initially developed under isohaline coordinates to track water masses and to compute estuarine exchange per range of salinity (MacCready, 2011). Flux computation over density layers has been used to quantify transport by internal waves from in situ data (Henderson, 2016; Inall et al., 2001) and numerical and in situ data (Spingys et al.,
2020).

  In the following, we detail the computation of the total volume flux $Q$, its Eulerian component $Q_{Eu}$ and the difference, which we term the Stokes drift volume flux $Q_r$. Finally, we detail the theoretical formulation of the Stokes drift for a mode-1 linear internal tide.

### 2.3.1 Total flux

The total volume flux for a time-varying section $h_T$ was computed using the following equation:

$$Q(t,T) = < \int_{h_T} u(z,t)\,dz > \tag{5}$$

Here $u$ is the unfiltered cross-shelf velocity and $u > 0$ corresponds to an onshore current. $h_T$ is the time-varying layer for which the unfiltered temperature increases by 1 °C. For example, $Q(t, 13°C)$ is the flux computed between 13 °C (strictly greater) and 14 °C (smaller or equal). Note that computing the flux within 2 °C is equivalent to the sum of 2 fluxes made

over a 1°C difference. The fluxes were then filtered using a 24 h low pass filter. Note that the filtering was applied after the flux computation so that $Q$ contains the net contribution of all the dynamics associated with temporal periods captured by the shortest sampling interval of the measurements, *i.e.,* 1 or 2 min (depending on the mooring, see section 2.1). The method differs to compute the Eulerian flux $Q_{Eu}$.

### 2.3.2    Eulerian flux computation

The Eulerian flux $Q_{Eu}$ was computed from the 24 h low pass filtered velocity and temperature field, *i.e.,* the background fields. This definition is equivalent to the Eulerian decomposition presented in (MacCready, 2011), as :

$$Q_{Eu}(t) = \int_{h_{<T>}^{Eu}} <u(z,t)> dz \qquad (6)$$

Here $<u>$ is the background velocity and $h_{<T>}^{Eu}$ is the layer for which the background (*i.e.,* low pass filtered) temperature $<T>$ increases by 1 °C (filtered velocity and temperature field are shown for each site in section 3). We evaluated the 24-

hour low-pass filter against the Demerliac tidal filter (Demerliac, 1974), applied to hourly data, and it was found to perform adequately (See Appendix Figure A3,A4,A5). The 24-hour low-pass filter was selected as it can be applied to non-hourly data and maintain the full temporal coverage of the measurements.

### 2.3.3    Uncertainty in flux estimates

We computed an estimate of the uncertainty in both the total and the Eulerian transport by propagating the measurement

uncertainties (see Supplementary material for details). The uncertainty was on the order of $10^{-2} m^2/s$. The uncertainty was generally small compared to the transport magnitude and only a small number of near-zero values had an uncertainty that was greater than 50% of the transport. However, removing these values resulted in time-averages that were slightly reduced, therefore we decided to not remove these values from the analysis to avoid introducing bias.

### 2.3.4    Stokes drift flux

We term the difference between the total and Eulerian fluxes ($Q$ and $Q_{Eu}$, respectively) the Stokes drift flux $Q_r$, see equation (7). This Stokes drift flux is the net volume flux resulting from processes such as barotropic tidal asymmetry, surface wave Stokes drift, internal tide Stokes drift and internal wave nonlinearity.

$$Q_r = Q - Q_{Eu} \qquad (7)$$

### 2.3.5    Theoretical Stokes volume flux of a linear mode-1 internal tide

The expression for the Stokes drift velocity for a mode-1 linear internal wave with a vertical structure of the vertical velocity $\phi_1(z)$ that propagates at a speed $c_{M2}$ is derived in (Franks et al., 2020) and first appears in appendix 6 in (Thorpe, 1968). The

derivation provided in (Franks et al., 2020) is recommended to the reader for its clarity. The Stokes drift velocity is:

$$u_S = \frac{A^2}{2} c_{M2} \left[ \left( \frac{\partial \phi_1}{\partial z} \right)^2 + \phi_1 \frac{\partial^2 \phi_1}{\partial z^2} \right] \tag{8}$$

We denote the analytical Stokes volume flux $Q_A$ and compute it over a layer derived from the background temperature field $h_{<T>}^{Eu}$ as in (Spingys et al., 2020) :

$$Q_A(t) = \int_{h_{<T>}^{Eu}} u_S dz \tag{9}$$

### 2.3.6 Internal tide amplitude

We computed the internal tide amplitude $A$ from the envelope of the band-pass filtered isotherm excursions (see next section for example). The envelope was taken as between the maximum and minimum excursion reached between the selected isotherms. The amplitude corresponds to half the distance between the minimum and the maximum of the envelope at each point in time. We selected isotherms from the mid water column, that continuously existed for the entire record and we considered a 1°C difference between the isotherms.

Because the stratification and depth were different at each site, the number of isotherms used varied. At N-BoB, the internal tide amplitude was computed using 2 isotherms and for the SE-BoB experiment we used 3 isotherms. We used a band-pass filter to retain the semi-diurnal oscillations. The low-pass filter at 2 h removed the NLIWs signature whereas the high-pass filter at 14 h removed the inertial oscillations. In the following, these filtered isotherms are referred to as tidal isotherms. The values of $c_{M2}$ and $\phi_1$ evolved in time and were computed following the methodology described in section 2.2.3.

## 3 Oceanographic conditions

Here, we focus on the temperature and velocity conditions at both sites to provide context for the analyses of the subtidal volume flux, the total cross-shelf volume flux and the analytical Stokes volume flux. We present the background field (low-pass filtered field), the amplitude, the vertical structure and the propagation speed of the mode 1 internal tide.

### 3.1 N-BoB

#### 3.1.1 Current and temperature fields

Compared to SE-BoB, velocity and temperature measurements at N-BoB were longer (1.5 months vs 1 month for SE-BoB) and later in Summer. Before September 28, the bottom temperature was less than 13°C and the near-surface temperature was above 17°C most of the time. Before September 28, the strongest background currents were in the middle of the water column (onshore) around 25 mab and near the surface (offshore) (Figure 2b). Near the seabed, the background currents were smaller than 0.1 m/s for most of the record (Figure 2b).

From September 07 to 12 the temperature was homogenized close to the seabed and close to the surface so that the stratification was slightly reduced while the barotropic tide increased from neap to spring tide (Figure 2a). From September 12 to

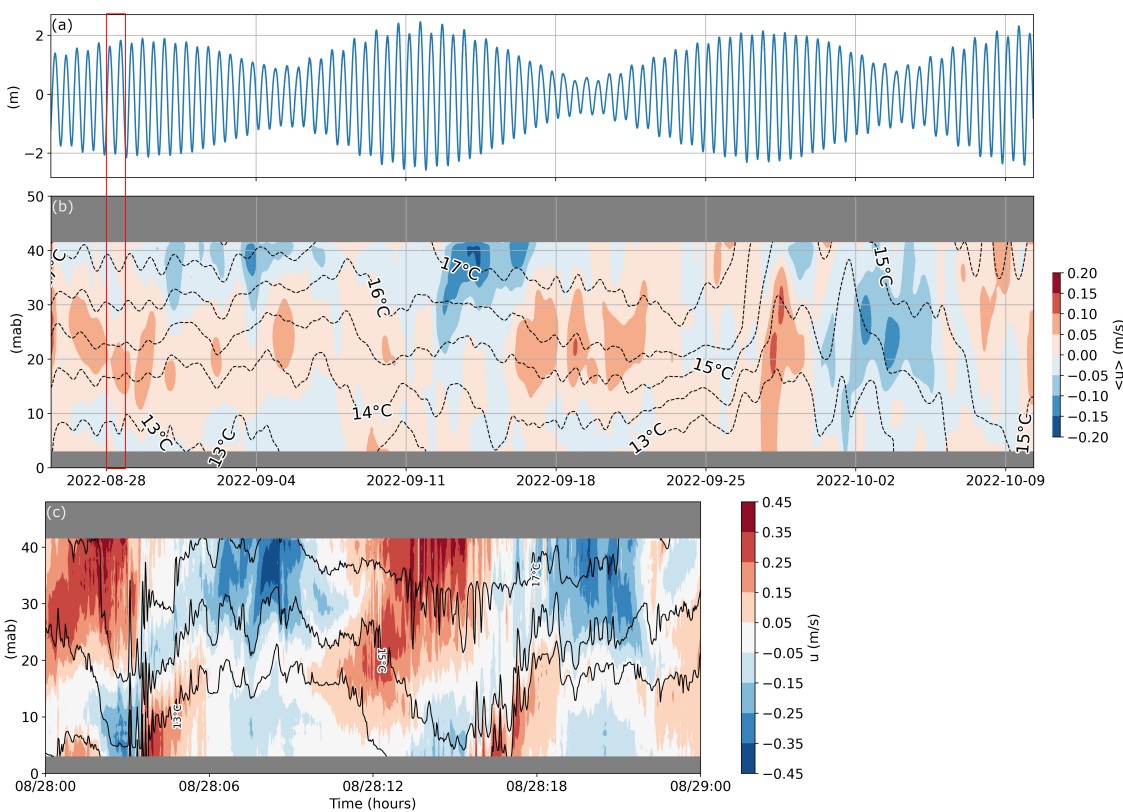

**Figure 2.** Oceanographic conditions at N-BoB, in 48 m of water depth. (a) Sea surface elevation anomaly over two months. (b) Background fields over two months. The colours are the 24h low pass filtered cross-shore velocity field $< u >$. The black dotted lines represent the 24-hour low pass filtered temperature evolution $< T >$. The definition of $< . >$ is given in 2.3.2. (c) Raw velocity and temperature field (labelled 13, 15, 17°C) for one day during August 25 (correspond to the red rectangle in (a) and (b)). Red is positive, onshore velocity. Blue is negative, offshore velocity.

29, during the neap tide (Figure 2a), the temperatures and the cross-shore currents were mostly constant (Figure 2b). Around September 29, a strong, nearly barotropic onshore current was observed, coinciding with ongoing homogenization of the water column (Figure 2b). After October 8, the temperature was almost constant in the whole water column (Figure 2b).

The raw (i.e. unfiltered) currents were much stronger, highlighting the role of high-frequency processes such as the tide and NLIWs (e.g., Figure 2c). The near-surface current reached up to 0.45 m/s, more than twice the maximum background velocity (Figure 2b.c). We regularly observed internal tide bores, NLIWs of elevation near the seabed and NLIWs of depression near the surface (Figure 2c).

### 3.1.2  Stratification and mode 1 internal tide

At N-BoB, the maximum displacement of isotherms was from -17 m to 7 m (Figure 3a). The associated internal tide amplitude

was around 12 m. At both sites, the internal tide displacements were not centred around 0.

During the first spring tide, between August 28 and September 01, the stratification was almost continuous with a slightly stronger vertical gradient close to the surface down to 15 mab which tended to favour NLIWs of depression (Figure 3b). The stratification changed from September 12 to 24, the warm surface layer thickened and the thermocline was between 20 and 15 mab. After September 17 the water destratified near the surface and a pycnocline formed around 15 mab. The associated

vertical structure of $\phi$ was maximum below the middle of the water column, favouring elevation NLIWs (Figure 3c.d).

From September 25 to the end of the record, the water column stratification decreased continuously. Around September 28, the internal tide amplitude was maintained between 5 m and 10 m. Before September 28 the isotherms 15°C and 16°C carried the largest amplitude internal tide. Just after September 28, the 14°C isotherm fell by 12 m and rose by 8 m (Figure 3a). Simultaneously, for the low-frequency signal, the 14°C isotherm rose with a steep slope (Figure 2b). This observation

suggested the superposition of an internal tide on a thermal front. After September 29, the stratification was weak (less than $10^{-4}s^{-2}$) and $c_{M2}$ dropped to 0.26 m.s$^{-1}$ (Figure 3b c). Around October 01 the internal tide displacement first decreased below 5 m, then amplified.

The internal tide amplitude decreased from September 04-11 to September 21-28 (Figure 3a). We note the asymmetry of the internal tide amplitude with a mean value less than 0 despite the filtering. The isotherm displacements were larger during

"low" internal tide (deepening of the isotherms) than during "high" internal tide. Two main reasons explain the asymmetric internal tide: first, the nonlinearity of the internal tide under asymmetrical stratification with a pycnocline close to the surface is expected to drive larger depression of isotherms compared with elevation. Second, packets of high-frequency negative spikes due to NLIWs may have also impacted the results of the filtering and led to a negative mean value of the filtered isotherm oscillations. Moreover, we observed steep slopes on the 15 and 16°C isotherms around September 28 (Figure 2b) which

suggest that the front may coincide with nonlinear internal waves. The asymmetry in isotherm displacement is expected to affect the vertical asymmetry of the transport. As discussed later, transport is enhanced close to the surface when isotherm depression is larger than elevation.

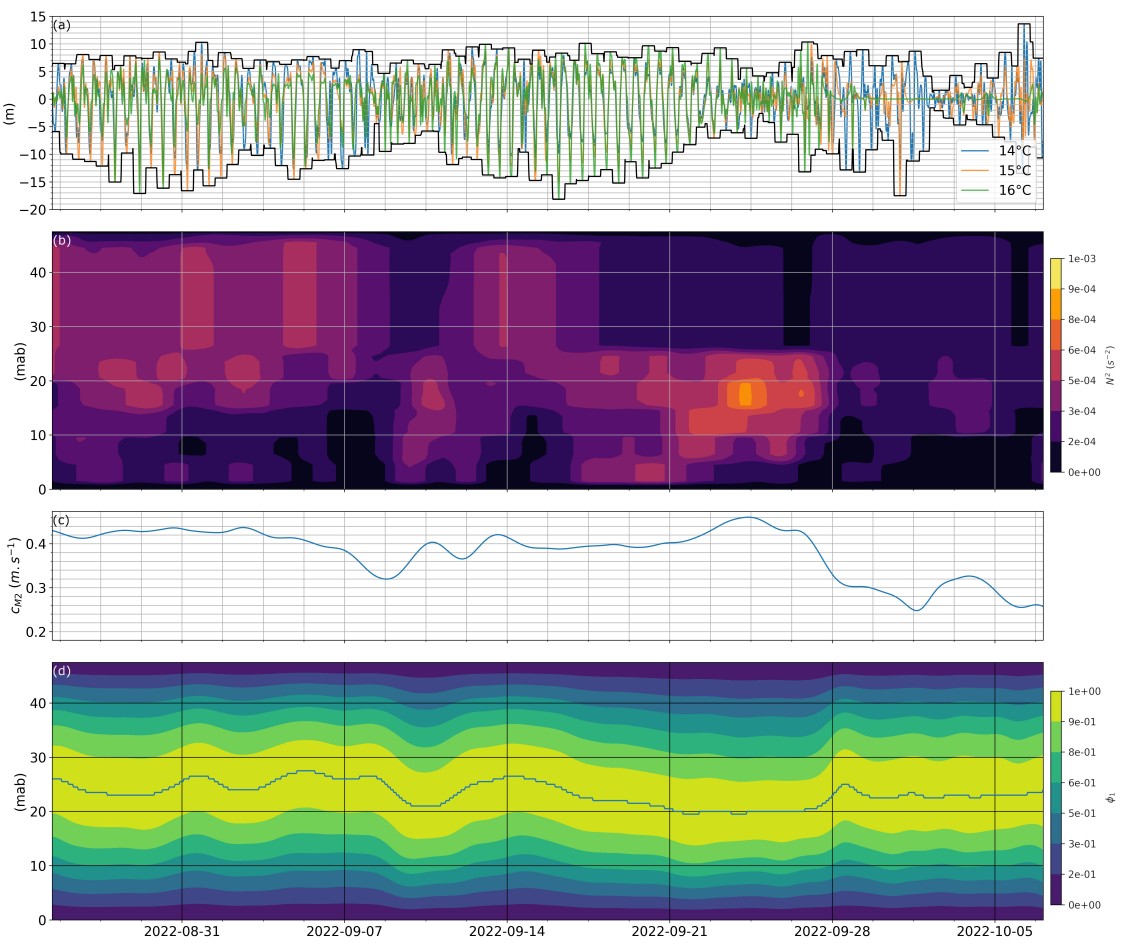

**Figure 3.** Internal tide characteristics at N-BoB. (a) 14°C, 15°C and 16°C isotherms, band-pass filtered between 2h and 14h. The black lines give the envelope and correspond to the internal tide amplitude $A$ as defined in 2.3.5. (b) Brünt Väisälä frequency defined in section 2.2.2 from 24 h low pass filtered data. (c) Mode 1 propagation speed $c_{M2}$ evolution. (d) $1^{st}$ vertical mode $\phi_1$ time evolution from equation (4). The continuous line indicates the position of maximal $\phi_1$.

## 3.2 SE-BoB

### 3.2.1 Current and temperature fields

The subtidal and total velocity did not show the same characteristics near the seabed. For most of the record, the subtidal current associated with temperatures below 14°C (around 5 mab) was offshore (Figure 4b). Evidence of the high-frequency peaks of onshore current and low temperatures (below 14°C) associated with NLIWs and the internal tide were present in the raw velocity record (one day example Figure 4c).

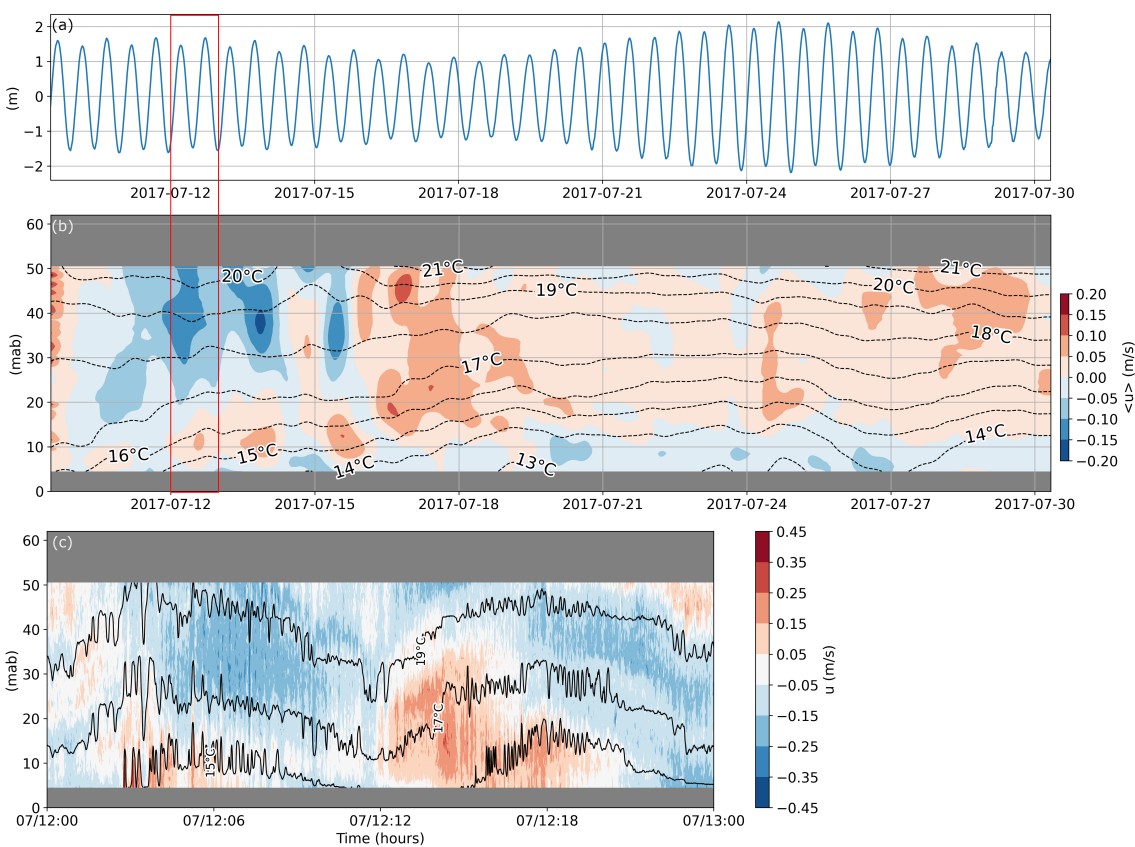

**Figure 4.** Oceanographic conditions at SE-BoB in 62 m water depth. (a) Sea surface elevation. (b) Background fields over one month. The colours are the 24 h low pass filtered cross-shore velocity field $<u>$. The black dotted lines represent the 24 h low pass filtered temperature evolution $<T>$. The definition of $<.>$ is given in 2.3.2. (c) Raw velocity and temperature field for the day of July 12 (corresponding to the red rectangle in (a) and (b)). Positive (onshore) velocity is indicated in red. Negative (offshore) velocity is indicated in blue.

For a detailed description of the dynamics, see (Moncuquet et al., 2025). At SE-BoB, the structure of the subtidal flow during the first spring tide was characteristic of upwelling (offshore surface layer flow and onshore bottom layer flow; (Lentz and Chapman, 2004)) from July 12 (Figure 4a.b). During the first spring tide, under upwelling conditions, the near-seabed (below 10 mab) temperature decreased. After July 15, during the neap tide, the near-seabed temperature kept decreasing and water masses of 13 °C were measured (Figure 4b). Just before July 27, downwelling dynamics generated a warming of the near seabed temperature during spring tides (Figure 4a.b).

### 3.2.2 Stratification and mode 1 internal tide

The conditions of stratification were suitable for both internal tide and NLIW propagation. An internal tide was observed for the whole record with an amplitude ranging from 4 m during neap tides to 8 m during spring tides (Figure 5a). The background

stratification was maintained during the whole record. Two pycnoclines whose intensity varied with time were separated by a less stratified region (Figure 5b). Initially, the stronger pycnocline was beneath 10 mab. It then rose around July 17 and weakened over time. On July 18 the near-surface pycnocline was the strongest. It is worth noting that the vertical resolution of our temperature sensors did not allow a precise description of the pycnocline position and width. The internal tide phase speed $c_{M2}$ increased from 0.40 m/s to 0.70 m/s, associated with the increase of the vertical averaged stratification later in the record (Figure 5c). The position where $\frac{d\phi}{dz} = 0$ was mostly around the middle of the water column and varied only slightly in time (Figure 5d).

A vertical mode maximum in the middle of the water column corresponds to a symmetrical stratification. Under symmetrical stratification, the usual KdV soliton is not expected. However, two pycnoclines (observed from July 18) can allow the formation of NLIWs of greater nonlinearity than the KdV soliton (Lamb, 2023). Moreover, the position and strength of the pycnocline are expected to modify the evolution of the internal tide along the shelf and the generation of NLIWs (Dauhajre et al., 2021).

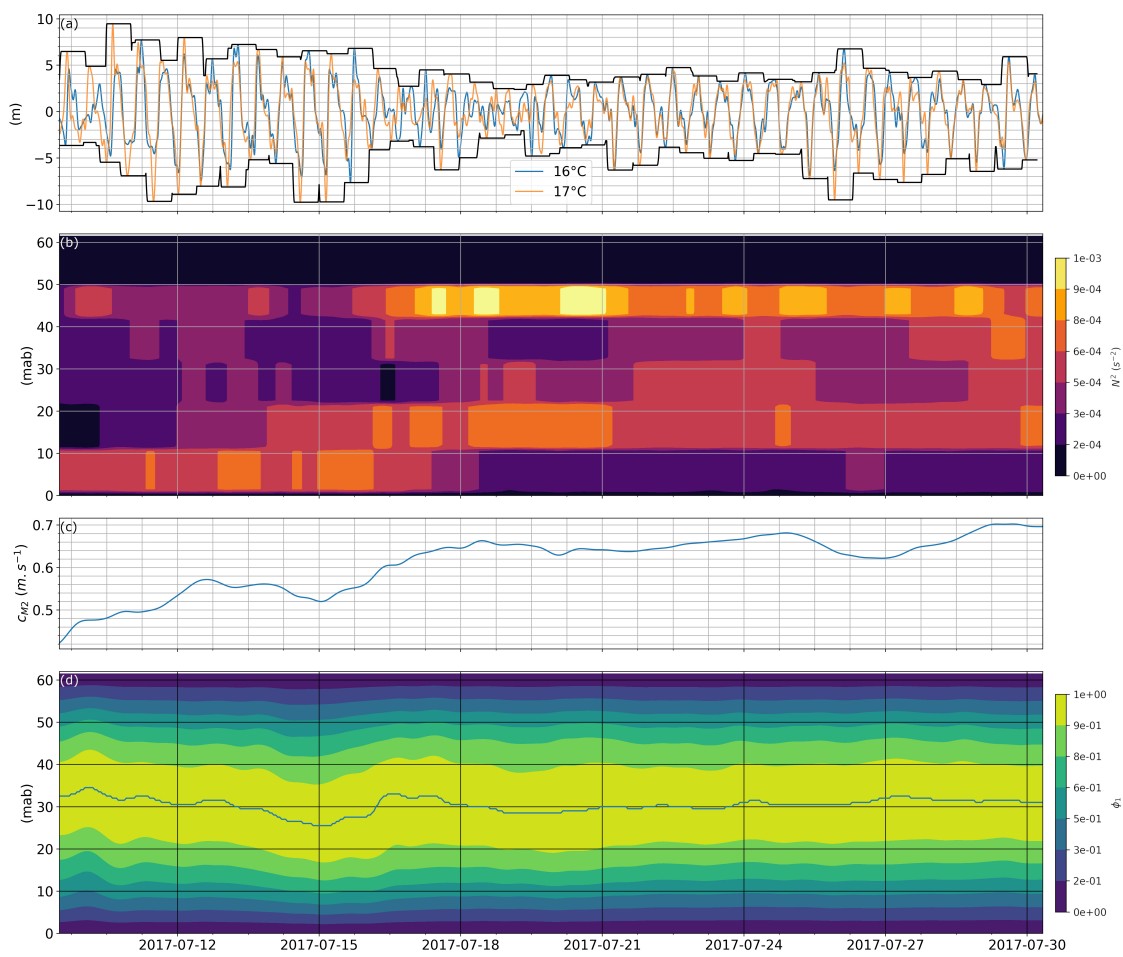

**Figure 5.** Internal tide characteristics at SE-BoB. (a) 16°C and 17°C tidal isotherms, band pass filtered between 2h and 14h. Only two isotherms are shown as we only selected isotherms measured for the whole period. The black lines give the envelope and correspond to the internal tide amplitude $A$ as defined in 2.3.5. (b) Brünt Väisälä frequency from 24 h low pass filtered data (from section 2.2.2) (c) Mode 1 propagation speed $c_{M2}$. (d) $1^{st}$ vertical mode $\phi_1$ from equation (4). The continuous line indicates the position of maximum $\phi_1$.

## 4 Cross-shelf transport diagnosed from moorings

In this section, we describe the cross-shelf volume fluxes per range of temperature at both SE-BoB and N-BoB. We compared the estimated Stokes drift volume flux $Q_r$ to the analytical Stokes volume flux $Q_A$ to determine the contribution of the linear internal tide Stokes drift to the observed Stokes drift volume flux. To determine which fluxes contribute to the total transport, we compared $Q$, $Q_{Eu}$ and $Q_r$. We first present the monthly averaged vertical profile of the fluxes. Then, we present the time series of the fluxes at both sites and investigate the impact of the background stratification variability.

## 4.1 Vertical profile of monthly averaged cross-shelf volume flux

For each layer, whose thickness is delimited by isotherms 1 °C apart, we calculated the total flux $Q$, the Eulerian flux $Q_{Eu}$, the Stokes drift flux $Q_r$ and the theoretical Stokes drift flux for linear internal waves $Q_A$, according to the formulas in section 2.3. The fluxes were computed for each temperature range and monthly averaged (Figure 6). The three-layer profiles of $Q_r$ were qualitatively similar for both sites and compared well with the Stokes drift flux expected for the linear internal tide $Q_A$ (Figures 6b.d). This result suggests that the time-averaged total Stokes drift flux was approximated to first order by the linear internal tide Stokes drift at both locations in summer over a month, suggesting the linear internal tide was the dominant contributor to the Stokes drift flux. It should be noted that the internal tide amplitude modulated the vertical profile of $Q_A$ homogeneously but not its vertical asymmetry which will be discussed in the following sections.

The main oceanographic processes that can significantly contribute to the Stokes drift flux are processes with periods smaller than 24 h which include surface tides, internal tides, internal waves, solitons, surface waves, storm surges (Huthnance, 1995; Huthnance et al., 2022). However, most of these processes are not expected to affect the results we present in this study. Storm surges are not expected to have a significant contribution at our sites during the summertime. Stokes drift due to surface waves is limited to the near surface, and we removed the measurements down to 10 m beneath the surface to limit their influence on the results. Moreover surface wave heights are small in the BoB during summer with mean significant wave height less than 2 m and mean period wave around 7 s (Charles et al., 2012). The barotropic tidal current is expected to generate a vertically homogeneous transport, except near the bottom boundary layer where the velocity decreases. Therefore the enhanced Stokes drift flux observed near the seabed is expected to be mainly driven by the internal tide and nonlinear internal waves.

### 4.1.1 Contribution of Stokes drift flux to the total volume flux

At N-BoB, the total flux $Q$ (Figure 6a) had a similar vertical profile to $Q_r$, *i.e.* onshore flux for minimum and maximum temperatures and offshore flux for medium temperatures (14 °C-16 °C). The Eulerian flux was weak and opposed the total flux, except for the coolest temperature class ($T <$13 °C) where both $Q$ and $Q_{Eu}$ were onshore. We observed the maximum Eulerian flux (0.1 m$^2$.s$^{-1}$) for medium temperatures, around 15 °C; $Q_{Eu}$ was small compared to the maximum $Q_r$. The absolute values of $Q_r$ were larger than $Q_{Eu}$. At N-BoB, the onshore flux by $Q_r$ integrated from 12°C to 14°C was around 0.25 m$^2$.s$^{-1}$. Hence, at N-BoB, the Eulerian flux was not the main contributor to the total flux $Q$ (Figure 6a). Instead, at N-BoB, the total cross-shore flux seemed to be mainly driven by the internal tide Stokes drift.

At SE-BoB, $Q$ also had a three-layer structure, well described by $Q_{Eu}$, apart for the coolest temperatures where it was similar to $Q_r$ (Figure 6c.d). $Q_{Eu}$ had a similar trend to $Q$ for 22 °C $\geq T >$ 14 °C with negligible values close to the surface (Figure 6c). In the interior, for temperatures between 19 °C and 15 °C, the total transport $Q$ changed signs and followed $Q_{Eu}$. Near the seabed, $Q$ and $Q_r$ were positive (onshore) for $T <$ 15°C which suggested a volume flux driven dominantly by internal waves. Near the surface, both $Q$ and $Q_r$ were positive (onshore) for $T >$ 20°C and less than 0.1 m$^2$.s$^{-1}$ (Figure 6c.d). At SE-BoB, the onshore volume flux of $Q_r$ from 12°C to 14°C was 0.25 m$^2$.s$^{-1}$.

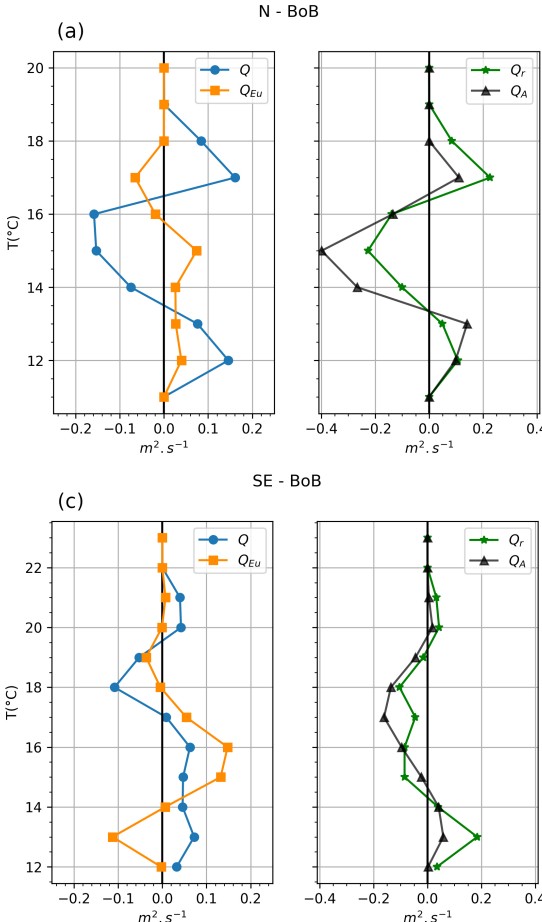

**Figure 6.** Monthly averaged cross-shore volume flux ($m^2.s^{-1}$) per temperature range. (a,b) N-BoB mooring. (c,d) SE-BoB mooring. Positive values are onshore, negative values are offshore. (a,c) $Q$ is the total flux (blue circles), $Q_{Eu}$ is the subtidal flux (orange rectangles) computed from the fields shown in Figure 2. (b,d) $Q_r$ is the difference between $Q$ and $Q_{Eu}$ (green stars). $Q_A$ is the Stokes drift for a linear internal tide (blue triangles).

At both sites, $Q_r$ close to the seabed was equivalent to the transport $q_w$ induced by weak upwelling forced by an along-shore wind speed $w$ of 4 $m.s^{-1}$ (using formula, $q_w = \frac{\rho_a C_D w^2}{\rho f}$ (Bowden, 1983) with a typical value of air drag coefficient $C_D$ of 1.4 $10.^{-3}$; air density $\rho_a$ of 1.22 kg.m$^{-3}$ and seawater density $\rho$ of 1025 kg.m$^{-3}$). Our results suggest that during the stratified period, at SE-BoB, the dynamics that drove cold water were induced by internal tides and NLIWs and lead to volume fluxes equivalent to a weak upwelling. A similar conclusion can be made at N-BoB for $Q_r$(17 $^oC < T^oC$). However it should be noted that part of the Eulerian fluxes may not have been observed because we removed measurements close to the seabed. Some low-frequency isotherm variability may have been present close to the seabed. The fluxes were exactly 0 for the coldest and warmest temperatures (Figure 6). Owing to the method, a flux that is exactly 0 means that the temperature

range was not measured. At SE-BoB, $Q_A(T \leq 12\ {}^oC) = 0$ (Figure 6d) because there are no temperatures below 12 °C at the
subtidal frequency (Figure 4). For the same reason $Q_{Eu}(T \leq 12^oC) = 0$ at SE-BoB (Figure 6c), whereas $Q(T \leq 12\ {}^oC) \neq 0$.
Therefore, $Q_r(T \leq 12\ {}^oC) \neq 0$, which means that 100 % of the onshore cross-shelf transport of water colder (or equal) to 12
°C was driven by tidal and high-frequency dynamics. Close to the seabed, Eulerian volume fluxes may partially cancel the
near-seabed upslope Stokes drift but they are not expected to be strong enough to change the sign of the total flux. Therefore
our results are expected to hold near the seabed and agree with previous studies that estimate net upslope volume fluxes near
the seabed due to internal tide and nonlinear internal wave Stokes drift in both a lake and in the ocean (Henderson, 2016;
Spingys et al., 2020).

### 4.1.2   The vertical profile of the Stokes drift volume flux

The Stokes velocity induced by a linear internal wave in a constant or symmetrical stratification profile is vertically sym-
metrical (Franks et al., 2020). A loss of symmetry in the stratification profile, *i.e.,* a near surface or near-seabed pycnocline,
translates into a loss of symmetry of the Stokes drift volume flux. At both sites, the Stokes drift volume flux $Q_A$ displayed
non-symmetrical flux as the stratification was asymmetrical.

    At both sites, the semi-analytical volume flux $Q_A$ was maximum (positive) near the seabed (*i.e.*, low temperature range)
(Figure 6b.d). At SE-BoB, the maximum $Q_A$ was $0.06\ m^2/s^1$ near the seabed, for temperatures 13 °C $\geq T >$ 12 °C. At N-BoB,
$Q_A$ has a near-seabed maximum of $0.18\ m^2/s^1$ for 14 °C $\geq T >$ 13 °C. The Stokes drift flux $Q_r$ was also asymmetrical but
did not always follow the same trend as $Q_A$. At SE-BoB, both $Q_r$ and $Q_A$ were maximum near the seabed, for temperatures
13 °C $\geq T >$ 12 °C. Therefore, the asymmetry of the SE-BoB Stokes drift flux can partly be explained by the asymmetric
stratification profile (Figure 6d). At N-BoB, the asymmetry of the analytical and computed volume flux were opposite. $Q_r$ was
greater near the surface, for a temperature around 17 °C, while $Q_A$ was maximum near the seabed for temperatures around
13 °C (Figure 6b). The stratification caused $Q_A$'s asymmetry, but could not explain the asymmetrical shape of $Q_r$. Moreover,
at both locations, the maximum positive values of Stokes drift flux were more than twice the maximum $Q_A$.

    Nonlinear processes are expected to enhance asymmetry in the vertical profile of the volume flux. NLIWs generate enhanced,
vertically-asymmetrical flux, therefore they are likely to drive this discrepancy. NLIWs generate a net horizontal flux in their
direction of propagation under the crest and in the opposite direction through the remainder of the water column (Shroyer et al.,
2010; Lamb, 1997; Zhang et al., 2018; Garwood et al., 2020). The loss of symmetry in the Stokes drift volume flux profile
may be a sign of the non-negligible impact of NLIWs. Asymmetry can also partly be due to the methodology as discussed
previously and high frequency surface waves. We acknowledge this potential contribution which we believe deserves further
investigation. The nonlinear internal tide is also expected to enhance vertical asymmetry in the Stokes drift volume flux. This
point will be further investigated by examining the time series of volume flux.

### 375   4.2   Time variability of the Stokes drift cross-shelf volume flux

In this section, we present the time variability of the four different cross-shelf volume flux estimates $Q$, $Q_A$, $Q_{Eu}$ and $Q_r$ at
both sites.

### 4.2.1  N-BoB - a site where the volume flux is dominantly driven by the Stokes drift flux

At N-BoB, under stratified conditions, the total volume flux $Q$ (Figure 7a) exhibited strong similarities with $Q_r$ (Figure 7c). Both fluxes were positive close to the surface and seabed, and negative for temperatures between 14°C and 16°C.

From August 27 to September 21, the vertical profile of the total flux could not be explained by the low-pass filtered Eulerian transport $Q_{Eu}$ and was therefore attributed to the Stokes drift flux (7a.b.c). Near the surface, for temperatures above 16°C, total subtidal flux was largest, positive and up to 1 $m^2.s^{-1}$ during the second spring tide (7a). The subtidal Eulerian flux did not display any positive values for temperatures above 16°C, partly because of the lack of temperature measurements near the surface (Figure 7b). The Stokes drift flux was also largest near the surface (Figure 7c). In the middle of the water column, the total flux was negative during the 1st spring tide for the temperature class between 14°C and 16°C, similar to the Stokes drift flux, with the same intensity between -0.3 $m^2.s^{-1}$ and -0.7 $m^2.s^{-1}$ (Figure 7a,c). For the same temperature class, $Q_A$ displayed a larger negative flux reaching -1.1 $m^2.s^{-1}$ (Figure 7d). During the 2nd spring tide the total offshore flux reached -1.1 $m^2.s^{-1}$ between 14°C and 18°C, which was mostly explained by the Eulerian flux (Figure 7a,c). Near the seabed (i.e for temperatures below 13°C), the total flux increased to 0.5 $m^2.s^{-1}$ during the first spring tide and 0.7 $m^2.s^{-1}$ during the second spring tide (Figure 7a). The subtidal Eulerian flux explained part of the near seabed flux during the first spring tide but was negligible during the second spring tide period.

The theoretical Stokes drift flux model was able to reproduce the vertical profile of $Q_r$ under the stratified conditions described in the previous section but with different intensity (Figure 7c.d). Near the surface, the theoretical Stokes drift flux $Q_A$ was minimal, with values around 0.1 $m^2.s^{-1}$, ten times smaller than the maximum values computed for $Q_r$. In the interior, $Q_A$ estimated a stronger offshore flux than what was observed in $Q_r$. Near the seabed, $Q_A$ and $Q_r$ exhibited similar values, reaching up to 0.5 $m^2.s^{-1}$ during the first spring tide and 0.7 $m^2.s^{-1}$ during the second spring tide. Finally, the temporal variability of $Q_r$ was greater than that of $Q_A$, revealing that the simple model of Stokes drift flux due to a linear internal tide could not capture all of the variability (Figure 7d).

Part of this discrepancy may be attributed to nonlinear internal waves. The impact of NLIWs was difficult to differentiate but we note that from August 27 to September 21, $Q_r$ reached values twice $Q_A$ for the near-surface temperature class (Figure 7c.d). At the same time, the offshore $Q_r$ was also enhanced in the interior (Figure 7c). Both depression and elevation NLIWs were regularly observed (Figure 7e). From September 14 to September 21, when the near-seabed $Q_r$ was maximum, we also observed a greater number of elevation NLIWs (Figure 7c.e).

After September 25, at the end of the record, the stratification was reduced and impacted both the total and Stokes drift subtidal flux (Figure 7). At the end of the record, we measured non-zero $Q$ only over a small range of temperatures as the water column became weakly stratified (Figure 7a). After September 28, the total flux $Q$ was unidirectional over the entire range of temperatures, similar to the Eulerian flux (Figure 7b). At the same time $Q_A$ dropped close to 0 because the amplitude of the internal tide dropped below 5 m. After that, both $Q_r$ and $Q_A$ became negligible (Figure 7c).

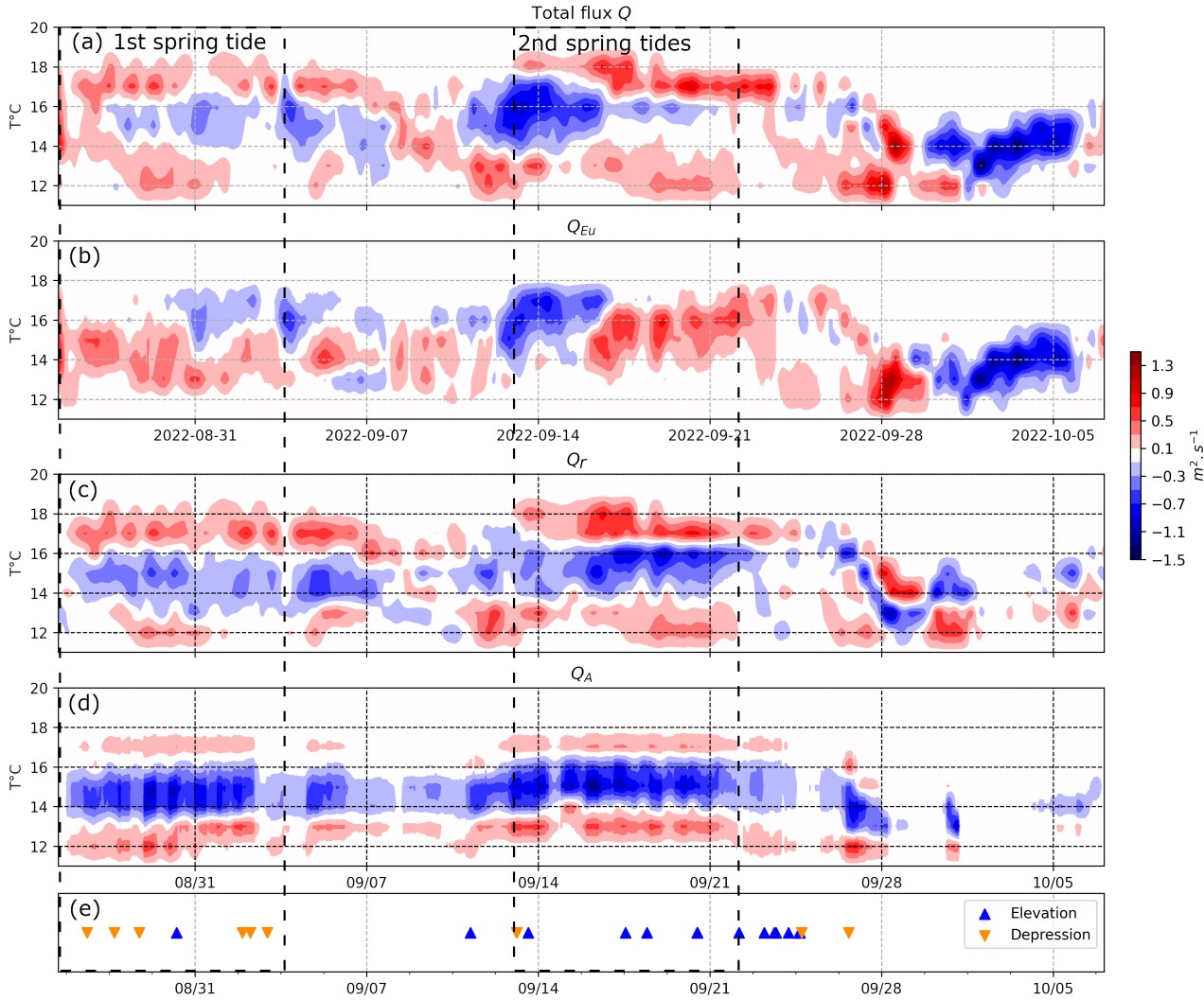

**Figure 7.** N-BoB fluxes and NLIWs from August 28 to October 06. Colours indicate the direction of volume flux. Red is onshore volume flux and blue offshore. (a) Total net cross-shelf volume flux $Q$. (b) Subtidal volume flux $Q_{Eu}$. (c) Stokes drift volume flux $Q_r$ (see definition in section 2.3). (d) Theoretical Stokes drift volume flux for a linear, M2, mode 1 internal tide $Q_A$ as defined in section 2.3. (e) NLIWs detection, blue triangles correspond to mode 1 NLIWs of elevation orange; downward triangles show mode 1 NLIWs of depression.

### 4.2.2  SE-BoB - a site where volume flux is mainly driven by subtidal variability

In contrast to N-BoB, the total subtidal volume flux $Q$ at SE-BoB did not have a clear three-layer structure throughout the entire time series (Figure 8a), even though the time-average $Q$ had a three-layer structure (Figure 6). Instead, $Q$ was similar to the subtidal Eulerian volume flux $Q_{Eu}$ (Figure 8b) except close to the seabed, for temperatures below 14 °C. For temperatures below 14 °C, $Q_{Eu}$ was offshore whereas $Q$ was onshore or nil. The Stokes drift $Q_r$ showed a constant onshore volume flux

near the seabed (Figure 8c). The linear internal tide Stokes drift had stronger onshore volume flux near the seabed compared to the surface but the magnitude was small compared to the other fluxes (Figure 8d). Therefore, at SE-BoB, the total volume flux was mainly subtidal except near the seabed where the Stokes drift flux counteracted the Eulerian volume flux, resulting in a weak onshore net volume flux.

The vertical profile of $Q_r$ was complex at the beginning of the record and was not always described as a three-layer profile. For example, during 7/12-13 the volume flux changed sign more than 3 times over the water column. During the time period between July 09 and July 15, we observed trains of NLIWs of elevation associated with internal tide bores (Figure 2a). The linear theoretical Stokes volume flux near the seabed was negligible during this period (Figure 8d). Nonlinearity from the internal tide and NLIWs were the likely drivers of $Q_r$ variation during the first spring tide.

After July 17, we no longer observed NLIW trains of elevation; instead, the elevation waves were generated by polarity reversal (Moncuquet et al., 2025). $Q_r$ was still stronger close to the seabed (Figure 8c). The near-absence of NLIWs after July 21 was associated with a Stokes drift flux that matched the theoretical linear tide Stokes drift volume flux, suggesting that the M2 mode 1 linear internal tide was driving the Stokes drift flux with enhancement due to internal tide steepening.

### 4.2.3 SE-BoB - interactions between internal waves and the wind-driven current

At SE-BoB we observed periods of time when a two layer structure $Q_{Eu}$ corresponding with stratified wind-driven upwelling (Batifoulier et al., 2012; Kersalé et al., 2016) coincided with a three-layer Stokes drift volume flux $Q_r$ similar to the theoretical internal tide Stokes drift structure.

During the 1st spring tide, between July 09 and July 16, we observed an offshore total volume flux $Q$ near the surface (warm temperatures) and an onshore flux in the bottom layer (cool temperatures) (Figure 9a), indicative of classical upwelling. However, for temperatures below 14 °C (near the seabed), $Q_{Eu}$ was actually negligible (Figure 9a). Under the stratified conditions, the onshore volume flux was further above the seabed, as described by (Lentz and Chapman, 2004). Lentz and Chapman (2004) presented observations made around the world, including negative volume flux near the seabed under upwelling on the Peru shelf — a location characterized by regular internal wave packets (Jackson et al., 2004). Note that the cold water was only present at tidal and higher frequencies and therefore was not retained in the low-pass filtered temperature field (Figure 4). Under the upwelling event, the Stokes drift flux drove the near-seabed onshore flux of cold water instead of the wind-driven upwelling, leading to net onshore near-seabed total flux. The onshore Stokes drift flux $Q_r$( 13°C $< T \leq$ 14°C) was up to 0.14 $m^2.s^{-1}$, and equivalent to the onshore subtidal flux $Q_{Eu}$( 14°C $< T \leq$ 15°C) (Figure 9a.b).

During the 2nd spring tide, the Eulerian volume flux at SE-BoB was onshore for warmer temperatures and offshore for colder temperatures (Figure 10a), which corresponded to a wind-driven downwelling structure. The total volume flux $Q$ was onshore over the entire temperature range (Figure 10a). The total near-surface and near-seabed volume flux were not explained by $Q_{Eu}$ and the near-seabed $Q_{Eu}$ was even offshore, with a maximum of 0.18 $m^2.s^{-1}$. This implies that the Stokes drift flux surpassed the offshore downwelling volume flux and drove an onshore total volume flux near the seabed.

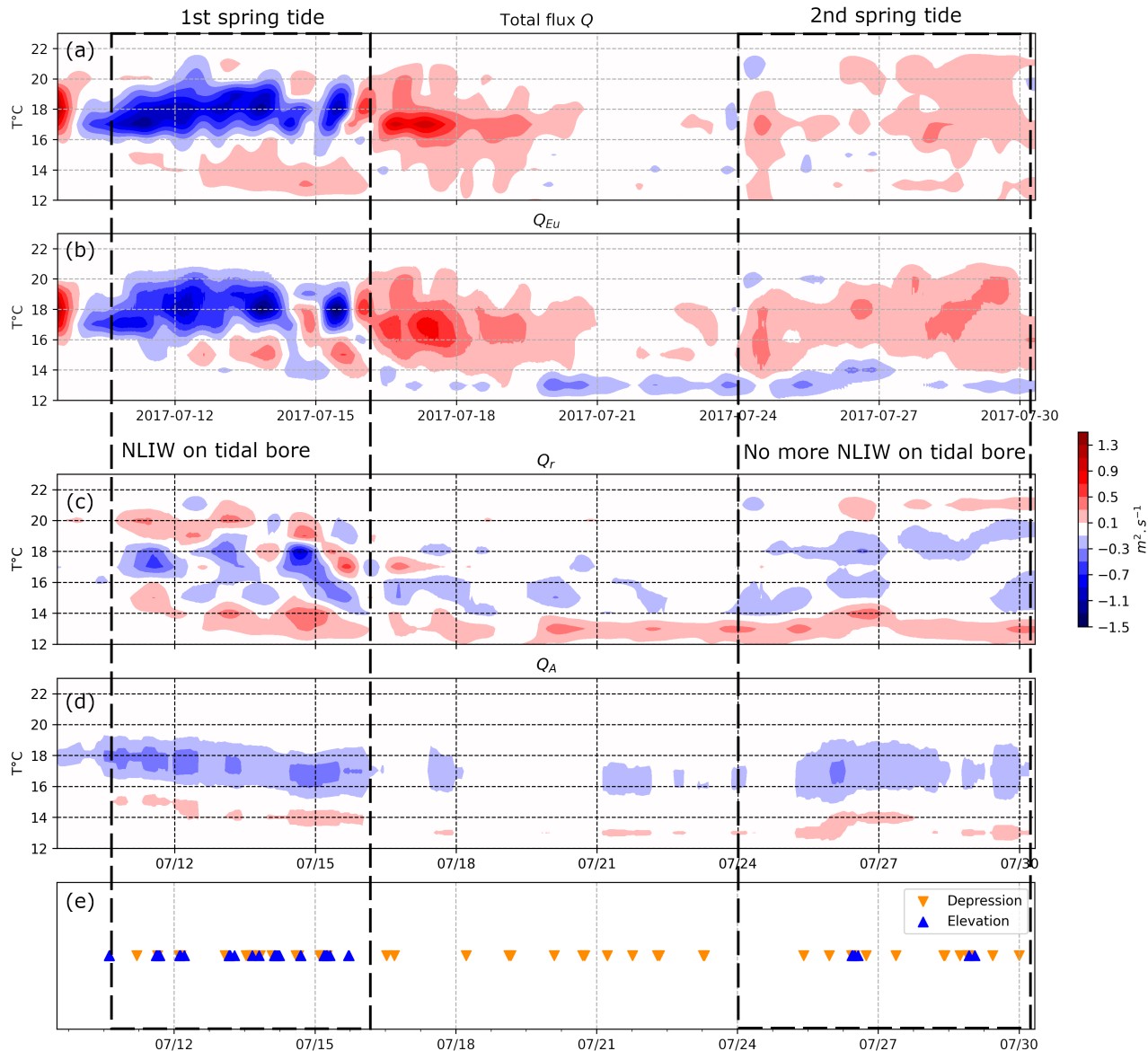

**Figure 8.** SE-BoB fluxes and NLIWs from July 09 to July 30. Colours indicate the direction of volume flux. Red is onshore volume flux and blue offshore. (a) Total net cross-shelf volume flux $Q$. (b) Subtidal volume flux $Q_{Eu}$. (c) Stokes drift volume flux $Q_r$ (see definition in section 2.3). (d) Theoretical Stokes volume flux for a linear, M2, mode 1 internal tide $Q_A$ as defined in section 2.3. (e) NLIWs, blue triangles correspond to mode 1 NLIWs of elevation, orange downward triangles show mode 1 NLIWs of depression.

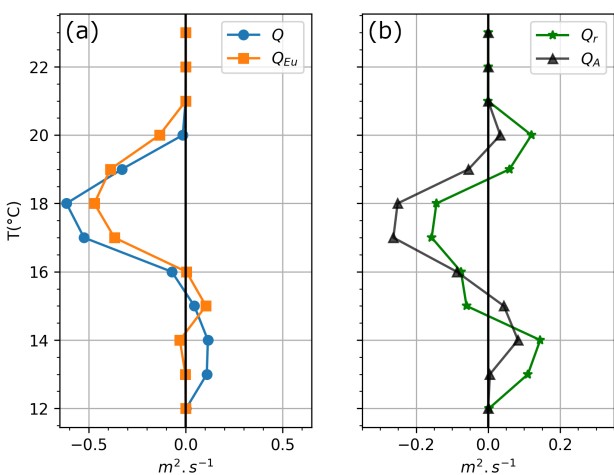

**Figure 9.** SE-BoB fluxes averaged over the $1^{st}$ spring cycle. (a) $Q$ is the net flux and $Q_{Eu}$ the Eulerian flux computed from the velocity field shown in the middle plot of Figure 2. (b) $Q_r$ is the Stokes drift flux, *i.e* difference between $Q$ and $Q_{Eu}$. $Q_A$ is the mean theoretical Stokes drift flux for a linear, M2 mode 1 internal tide.

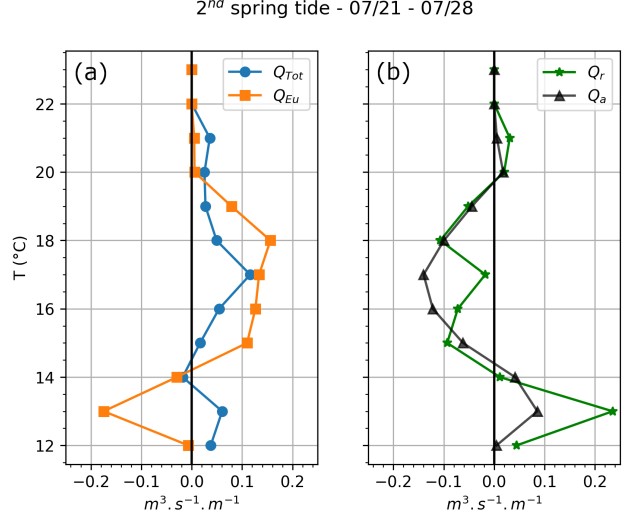

**Figure 10.** SE-BoB fluxes averaged over the $2^{nd}$ spring cycle. (a) $Q$ is the total flux and $Q_{Eu}$ the Eulerian flux computed from the velocity field shown in the middle plot in Figure 2. (b) $Q_r$ is the Stokes drift volume flux, *i.e.,* difference between $Q$ and $Q_{Eu}$. $Q_A$ is the theoretical mean Stokes drift volume flux for a linear M2 mode 1 internal tide.

## 5  Discussion and Conclusions

In this study, we quantified the total, Eulerian and Stokes drift cross-shelf subtidal volume flux at two distant coastal sites in the Bay of Biscay. Instrument uncertainty propagation analysis showed that uncertainty was generally small compared to the

transport magnitude ($10^{-2} m^2/s$) and only a small number of near zero values had an uncertainty that was greater than 50% of the transport. This demonstrates the robustness of both the residual transport structure and magnitude (see Supplementary materials). At the two sites, the month-averaged Stokes drift flux (difference between the total and Eulerian fluxes), matched well the theoretical profile of the internal wave Stokes drift for a mode 1 linear internal tide. Therefore, we attributed the Stokes drift flux at both study sites to be mainly driven by the internal tide. The time series of the Stokes drift fluxes revealed the additional likely important contribution of nonlinear internal waves at these sites, thereby we attribute the Stokes drift flux to internal wave pumping due to the linear internal tide, internal tide steepening and nonlinear internal waves. The IWP was modulated by background stratification, interaction with wind-driven currents, and the presence of nonlinear processes. At N-BoB, Eulerian transport was weak and IWP drove the total flux vertical structure. At SE-BoB, the total flux was dominated by wind-driven currents and modulated by IWP, apart from near the seabed where IWP dominated the onshore flux.

Onshore near-seabed volume flux was dominated by IWP at both sites. At N-BoB, the monthly-averaged volume flux due to IWP was 0.25 $m^2.s^{-1}$ near the seabed, -0.5 $m^2.s^{-1}$ in the interior and, 0.27 $m^2.s^{-1}$ near the surface, which is comparable to previous values observed in the Celtic Sea by Huthnance et al. (2022). On the Armorican shelf, at N-BoB, the total volume flux was predominantly driven by the IWP during the stratified period and spring tides. The theoretical Stokes drift internal tide volume flux represented the observed vertical profile of the IWP, but not the magnitude and asymmetry, suggesting the important impact of nonlinear processes such as internal tide steepening and NLIWs on the Stokes drift volume flux. At SE-BoB, the monthly-averaged volume flux was 0.25 $m^2.s^{-1}$ near the seabed, -0.33 $m^2.s^{-1}$ in the interior, and less than 0.1 $m^2.s^{-1}$ near the surface. Under both upwelling and downwelling events, the IWP drove an onshore near-seabed volume flux for the entire record, similar to the 0.3 $m^2.s^{-1}$ sustained near-seabed volume flux by NLIWs observed by Inall et al. (2001) on the Malin shelf.

Nonlinearity from internal tide steepening, NLIWs, and the interaction between internal waves and mesoscale dynamics likely explain the discrepancy between the IWP vertical profile and the Stokes drift theoretical internal tide profile. At both sites, nonlinearity modulated the Stokes drift volume flux. The theoretical Stokes drift for a M2 linear mode 1 internal tide explained only part of the Stokes drift flux profile and magnitude throughout much of the time series. NLIWs were shown to account for some of the discrepancies between the theoretical and observed Stokes drift. Additional moored observations that sample a wider variety of NLIWs are required to more definitively attribute Stokes drift volume flux to NLIWs.

The uncertainty in transport is expected to be enhanced by the impact of the NLIW on measurements. The lagged arrival times of short-wavelength signals at different beams of an ADCP can cause substantial differences in measured velocities. This is due to the beam-to-Earth coordinate transformation (Scotti et al., 2005). To correct this effect, the ADCP must be in a specific beam configuration. This was not the case for the ADCP used at the SE-BoB. Furthermore, the velocities must be corrected for each wave, taking into account their specific propagation speed and direction. The correction is not possible when waves superpose. Currently, the effect of NLIWs on the different components of the transport cannot be assessed. A study focusing on NLIW transport and its contribution to the total Eulerian and residual transport would clarify their impact on cross-shelf transport. The surface tide and high frequency surface waves could also contribute to the observed Stokes drift. However, we

estimate these contributions should be small during the observation period due to the weak surface tide asymmetry in the region and the weak high frequency surface waves.

The impact of nonlinear internal tides and NLIWs could be further investigated using a realistic high-resolution hydrostatic numerical configuration of the BoB to compare cross-shelf flux under stratified and non-stratified conditions and to further diagnose the nonlinearity of the internal tide on the shelf. The relative contribution of NLIWs and internal tides to the volume flux should be further investigated using a non-hydrostatic model.

Using a numerical model, Bourgault et al. (2014) demonstrated the role of NLIWs in transporting resuspended sediment shoreward. This led Cheriton et al. (2014) to suggest that midshelf mud belts could be partially controlled by internal waves, via erosion of mud at the outer-shelf being transported towards the midshelf. Similarly, the onshore near-seabed transport observed at N-BoB could supply the BoB midshelf mud belt (the "Grande Vasière"). A deeper investigation of the internal wave transport on the BoB shelf could help explain the unique sediment distribution across the shelf. At the SE-BoB, Cirac et al. (2000) observed the absence of sand and unique and unexplainable sedimentary patterns around 80 m water depth. We suggest that the internal tide and waves could potentially impact the sediment distribution across the SE-BoB by inducing dominant near-seabed onshore transport.

At the Northern Bay of Biscay, close to N-BoB, Laes et al. (2007) made vertical measurements of iron concentration over the shelf. They observed enhanced values of iron close to the seabed, which could not be explained by wind-driven mechanisms such as upwelling. Instead, the authors hypothesized that internal waves may drive near-seabed iron transport. This finding is supported by our observations. A combination of biochemical and physical measurements on the shelf is needed to better explain the transport of nutrients such as iron over the shelf.

## Appendix A

### A1

Density was computed as a function of the temperature using the linear regression observed from the MVP measurements. The relationship between density and conductivity was considered stationary.

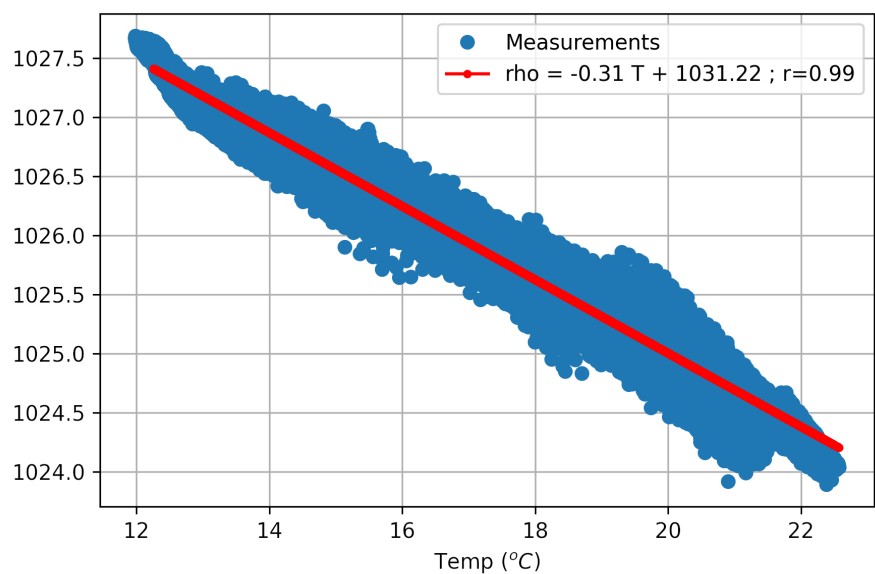

**Figure A1.** SE-BoB density-temperature relationship. The density is given as a function of the temperature measured from 3 days CTD measurements (corresponds to 8 transects across the shelf) mounted on the moving vessel profiler (MVP). The red line corresponds to the linear regression with the given coefficient.

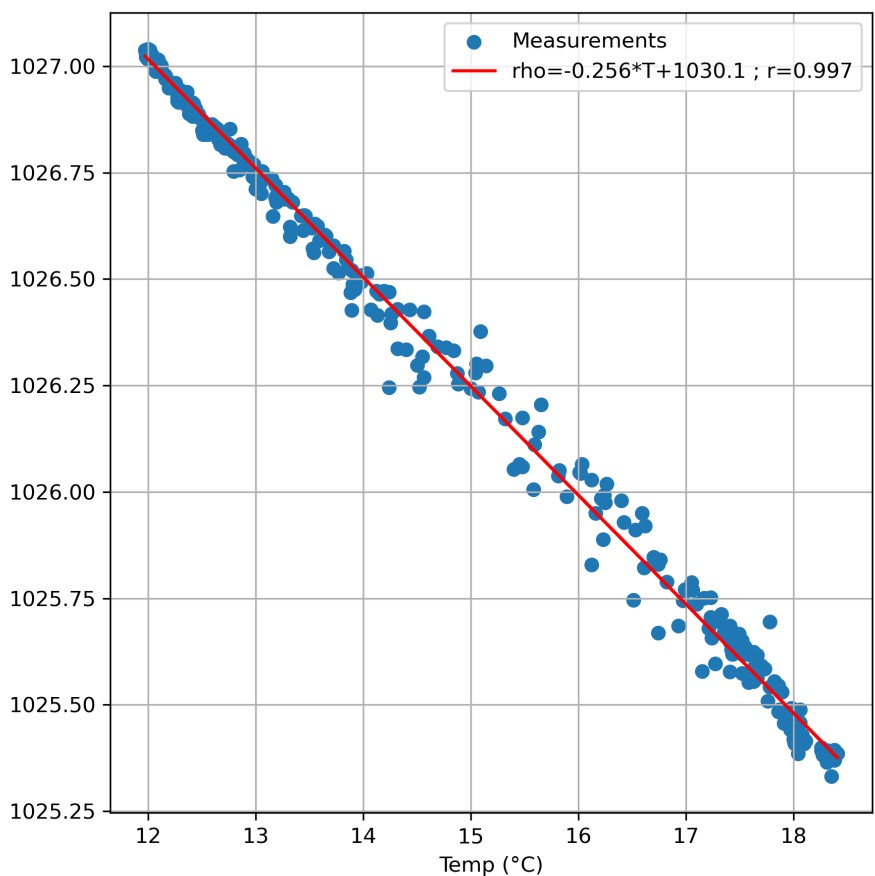

**Figure A2.** N-BoB density-temperature relationship. The density is given as a function of the temperature measured from CTD measurements mounted on the moving vessel profiler (MVP). The red line corresponds to the linear regression with the given coefficient.

**A2**

Eulerian transport was computed using a 24-hour low-pass filter and Demerliac tidal filtering applied to hourly data. The time-averaged $Q_{(Eu)}$ was similar with both methods. Although the filtered Demerliac data display less variability, the structure and intensity of the observed events are similar to those in the 24-hour low-pass dataset, which was considered adequate for the current analysis.

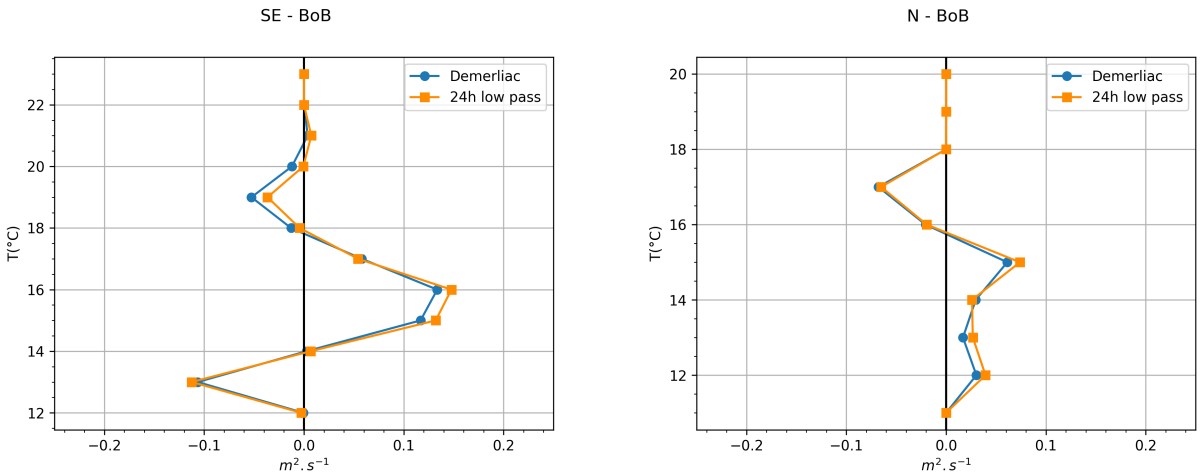

**Figure A3.** Time averaged Eulerian transport $Q_E u$ using a 24h low pass filter and the tidal filter Demerliac. (SE-BoB on the left and N-BoB on the right.)

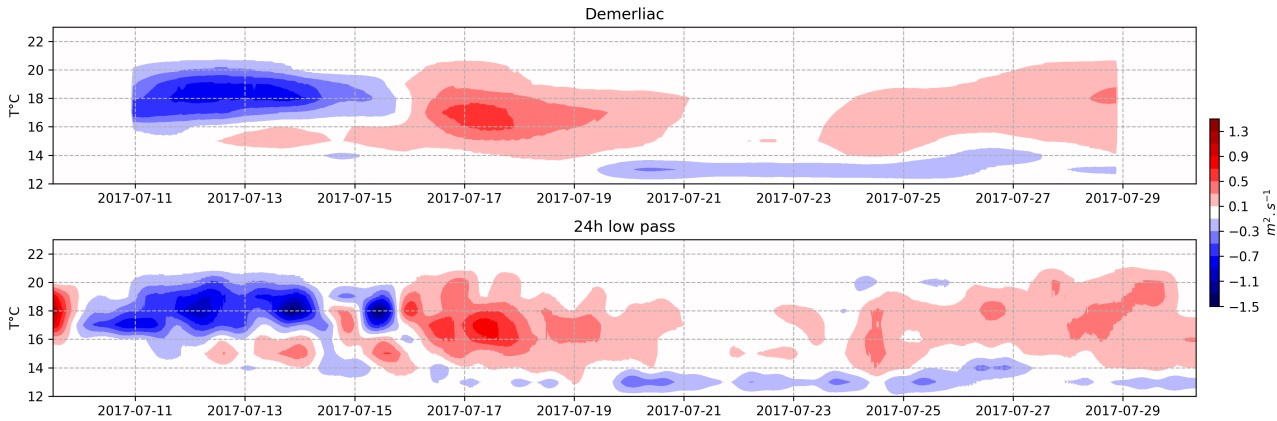

**Figure A4.** The time series of the Eulerian transport $Q_E u$ using a Demerliac filter (top) and a 24h low pass filter (bottom) at the SE-BoB.

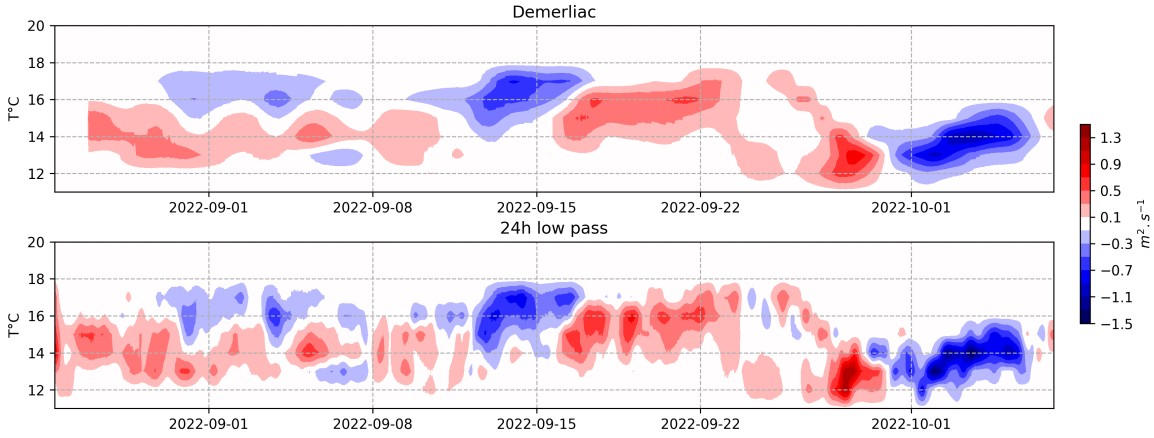

**Figure A5.** The time series of the Eulerian transport $Q_E u$ using a Demerliac filter (top) and a 24h low pass filter (bottom) at the N-BoB.

*Data availability.* SE-BoB : LAZURE Pascal, PUILLAT Ingrid (2017) ETOILE cruise, RV Côtes De La Manche, https://doi.org/10.17600/17010800

N-BoB : LAZURE Pascal (2022) SOLIBOB 2022 cruise, RV Côtes De La Manche, https://doi.org/10.17600/18002070

*Author contributions.* – A.M : Conceived and designed the analysis; Performed the analysis; Wrote the Paper

– N.J : Conceived and designed the analysis; Wrote the Paper

– L.B : Conceived and designed the analysis; Wrote the Paper

– F.D : Collected the data; Conceived and designed the analysis; Wrote the Paper

– P.L : Conceived and designed the analysis; Collected the data; Contributed data or analysis tools; Wrote the paper

*Competing interests.* No competing interests are present

*Acknowledgements.* We are grateful for the support of the ISblue project, Interdisciplinary graduate school for the blue planet (ANR-17-EURE-0015) and co-funded by a grant from the French government under the program "Investissements d'Avenir" embedded in France 2030 whose efforts made this research possible. We acknowledge Ifremer PhD scholarship program for the additional funding during A. Moncuquet

PhD. This study was partly funded by the Laboratory for Ocean Physics and Satellite remote sensing and Ifremer. Jones was supported by the Australian Research Council (Grants IH200100009, and DP210102745). The ETOILE oceanographic campaign has received funding from the European Commission's Horizon 2020 Research and Innovation program (H2020 JERICO-NEXT). The authors would also like to thank the RV Côte de la Manche crew for their dedicated work as well as the technical team from Ifremer and SHOM for their technical assistance.

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
