# Peer review of "Observations of cross-shelf transport due to internal wave pumping on the Bay of Biscay shelf"

_EGUsphere, 2025_

## Author Response (AR1)

**Comments on egusphere – 2025-2072 Anonymous Referee #1**

(1) comments from referees
(2) author response
(3) author's changes in manuscript.

(1) This paper applies a volume transport decomposition to identify the relative importance of Eulerian and Stokes' transports to the next cross shelf transport in the Bay of Biscay. They focus on the role of the Stokes' transport by the linear internal tide and demonstrate that the derived transports fit well with theoretical understanding and are the dominant driver of cross shelf transport for some locations and times. Additionally, they highlight times when the transport is significantly modified by non-linearity. The transport across the shelf edge is an important problem and the role of the linear internal tide is not well understood with very few (only 2 others to my knowledge) observational studies. The paper is well written and clear with results that will be of interest to the readers of Ocean Science, as a result I recommend publication of the manuscript. I have made a few minor suggestions below that the authors may wish to consider.

(2) Thank you for reviewing our manuscript. We appreciate your precise feedback.

(1) Fig 1. It would be interesting to see the mooring derived profiles alongside some higher resolution ship profiles if you have any (e.g. from the MVP) to understand how well the resolution of the thermistors is capturing the stratification. From the figure I expect it is doing a good job but would be great to check.

(2) We thank the referee for this comment. First, we would like to highlight that Figure 1 provides an initial overview of the mean temperature profile. This averaged temperature profile was not used in the computations. We have previously compared the MVP to the high temporal resolution mooring data in Figure 7 from the following paper: https://agupubs.onlinelibrary.wiley.com/doi/10.1029/2024JC021021. Here we present the evolution of the temperature field over the course of a day, using both MVP measurements and mooring data. The comparison demonstrates that the mooring data has adequate vertical resolution.

(1) Line 197 – I agree that the Franks derivation is clear however it is far from the first paper to show this result (as acknowledged by the authors). I think it first appears in an appendix of Thorpe 1968, so I would reference that here also. Thorpe, S.A. 1968. On the shape of progressive internal waves. Philosophical Transactions of the Royal Society A 263(1145):563–614, https://doi.org/ 10.1098/rsta.1968.0033.

(2) We have now added the original reference.

(3) Author changes in line 197.

(1) Fig 2c – The caption lists different isotherms to the ones labelled in the figure (the caption says 14,16,18,20 and the figure shows 13,15,17).

(2) Thank you for catching this error it has now been corrected. Following the second referee comments Figure 2 has been upgraded and now the isotherms shown are 15, 17 and 19.

(3) Author changed Fig 2c and its caption.

(1) Line 319 – Typo Qr1

(2) The typo has been modified. Thank you.

(3) See the change in line 318.

**Comments on egusphere – 2025-2072 Anonymour Referee #2**

(1) comments from referees
(2) author response
(3) author's changes in manuscript.

(1) The authors study observed cross-shelf volume transports associated with mode-1 internal tides and high-frequency Nonlinear Internal Waves (NLIWs) in two contrasting mid-shelf locations. The vertical structure of the Stokes drift profiles Qr is a particular focus, with monthly-averaged volume transport estimates of ~0.1-0.2 m2/s in different parts of the water column. Comparison with theoretical predictions for the shape of Qr suggest this is a significant cross-shelf transport mechanism in the Armorican and Aquitaine shelves. Interaction with the wind-driven upwelling/downwelling Eulerian circulation is also discussed.

The text and figures are mostly clear, though figure labels are sometimes hard to read. I appreciate the comparison between the two different regimes and the Stokes volume transport estimates, since this is a difficult process to measure, and is not present in most full-complexity numerical models.

I see some major and minor issues with the manuscript in its present form, which I detail below. Briefly, I feel that a couple of sources of instrumental error (random instrumental noise and the assumption of flow homogeneity across the ADCPs' beam separation) need to be examined to add more confidence to the results.

(2) Thank you for your efforts in reviewing our manuscript. We are also grateful for the discussion you have initiated.

**Major**

(1) M1: Since the transports Q, QEu and Qr (and other quantitative estimates) are based on measurements which contain random error, it is important to have error bounds derived from the propagation of the instrumental noise. This would add confidence to the results, for example the comparison between experimental and theoretical Stokes drift fluxes in Section 4.1. Figures 1(c,e), 6, 9, and 10 would also benefit from this analysis.

(2) See the "M1.pdf" file linked to the discussion.

(1) M2: The lagged arrival times of short-wavelength signals at different beams of an ADCP can cause substantial differences in measured velocities due to the beam-to-Earth coordinate transformation. In this dataset, the presence of NLIWs and tidal bores can add spurious transport to the raw time series. Scotti et al. (2005, https://doi.org/10.1175/JTECH1731.1) propose a method to correct for these efects, which I think is necessary here. This would eliminate potentially spurious NLIWs and tidal bore signals in the raw velocity records that could affect the total Q estimates (and hence Qr). This could change the interpretation of the Qr profiles' shapes, e.g., the discussion in lines 449-463.

(2) Reconstructing a coherent velocity time series that would be corrected under each appearance of NLIWs is not technically possible (at least with the dataset we are using). This is due to several limitations:

- Only one of the two ADCPs used in this paper is in beam coordinates. The Scotti et al. (2005) correction is not possible for ADCP recording data in Earth coordinates.

- The methodology given in Scotti et al. (2005) requires a strong backscatter signal associated to the NLIW and this is not always the case in our ADCP dataset. Retrieving the propagation speed and direction is therefore not possible for all the NLIW events.

- The methodology was initially developed to correct the velocity under a train of NLIWs with constant propagation speed and direction. When waves superpose the methodology is not applicable. This happens multiple times in our datasets, either with the superposition of elevation and depression waves, or waves propagating independently over a short time period. Correcting the velocity under hundreds of NLIWs with varying propagation speed and direction, and reconstructing a coherent

time series (i.e. without adding uncontrolled time dependant errors ) seems unrealistic.

(3) We acknowledge that the lagged arrival times of short-wavelength signals at different beams of an ADCP will add uncertainties in the measured velocity. The NLIWs are expected to increase the measurement uncertainty. To better clarify this point we propose to add the following paragraph in the discussion section,  line 449-463 :

"The uncertainty in transport is expected to be enhanced by the impact of the NLIW on measurements. The lagged arrival times of short-wavelength signals at different beams of an ADCP can cause substantial differences in measured velocities. This is due to the beam-to-Earth coordinate transformation (Scotti et al. (2005, [https://doi.org/10.1175/JTECH1731.1](https://doi.org/10.1175/JTECH1731.1) ). To correct this effect, the ADCP must be in a specific beam configuration. This was not the case for the ADCP used at the SE-BoB. Furthermore, the velocities must be corrected for each wave, taking into account their specific propagation speed and direction. The correction is not possible when waves superpose. Currently, the effect of NLIW on the different component of the transport cannot be assessed. "

(1) M3 (Line 132): It is worth checking that the 24 h cutoff period effectively filters all major tidal constituents. I wonder if it might leak some M2 variance, as M2 is the most energetic constituent and has a period that is not a multiple of 24 h.

(2) We computed the Eulerian transport using the DEMERLIAC tidal filter, which is effectively used by SHOM. We compared these results with the 24-hour low-pass filter results presented in the referenced paper.

The time-averaged profiles of Eulerian transport are equivalent at both sites using either method (Figure 1). The time series differ. The 24-hour low-pass filter is less effective at removing tidal constituents but The DEMERLIAC filter operates on hourly data which averaged out the contribution long term contribution of the high frequency dynamics such as the NLIW. Additionally, the tidal filter operates on time windows of several days, causing the final results to lack the first and last days of the record (Figures 2 & 3).

The structures described in the paper remain present in both results:

- At N-BoB: The 3-layer structure is observed in both results on 2022-09-07. The destruction of the stratification is observed in both results. (Figure 2)
- At SE-BoB: The DEMERLIAC filter removes the complex 3-layer structure on 2017-07-15 and reduces negative Eulerian transport. However, key structures

remain in both results, including large negative transport at the beginning of the record and negative transport near the bottom after 2017-07-21 (Figure 3)

In conclusion we kept the 24h low pass filter after verifying that the monthly averaged profile was equivalent and the main structures in the time series were equivalent to a specific tidal filter DEMERLIAC. The 24h low pass filter was preferred because it can be applied to high-frequency data and it does not remove the first and last day of measurements.

[Figure]

Figure 1 : time averaged eulerian transport Q_Eu using a 24h low pass filter and the tidal filter DEMERLIAC. (N-BoB on the left and SE-BoB on the right.

[Figure]

Figure 2 : time series of the eulerian transport Q_Eu using a Demerliac filter (top) and a 24h low pass filter (bottom) at the N-BoB.

[Figure]

Figure 3 : time series of the eulerian transport Q_Eu using a Demerliac filter (top) and a 24h low pass filter (bottom) at the SE-BoB.

(1) M4 (Lines 192-193, 301): I understand the Stokes drift from the other processes listed, but why would density-driven mean flows have a Stokes component? It is not an oscillatory process.

(2) We acknowledge that our previous statement suggesting a possible contribution of density-driven currents to the Stokes drift was incorrect. As clarified in the Generalized Lagrangian Mean framework (Andrews & McIntyre, 1978; Bühler, 2014), Stokes drift arises only in the presence of wave-like oscillatory motions. We have corrected the manuscript accordingly.

**References**

- Andrews, D.G. & McIntyre, M.E. (1978). *An exact theory of nonlinear waves on a Lagrangian-mean flow*. **Journal of Fluid Mechanics**, 89(4), 609–646.
- Bühler, O. (2014). *Waves and Mean Flows*. Cambridge University Press.

(3) Mention to density-driven flows removed from lines 194, 303 and 305.

**Minor points:**

(1) m1 (line 54): "The southern shelf of the BoB presents the smallest barotropic tide of the shelf". This statement could use a reference.

(3) Author added citation line 54.

(1) m2 (Line 79): "on average the regional circulation is less than 2.5 cm/s". Is this a typical along-shelf or cross-shelf velocity, or a speed?

(2) Speed

(3) Modified line 79

(1) m3 (96-97): It would be helpful to include the temperature and pressure sensors' specifications such as manufacturer and model.

(2) – (3) Ok, the sensors specification have been added line 97 and line 120.

(1) m4 (Line 98-99): What is the source for the 0.15 degrees C accuracy of the temperature sensor?

(2) The temperature sensor calibration were documented in section 3 in (Lazure et al. 2019 https://archimer.ifremer.fr/doc/00630/74245/) . The Mastodon moorings were the one used in both dataset (SE-BoB et N-BoB). The uncertainty of the temperature sensor is actually 0.1°C.

(3) We added the refence in line 99 and line 123.

(1) m5 (Line 100-101): Same as in m4, it would be good to include the ADCP's single-ping standard deviation for error propagation.

(2) SE-BoB site : 8cm/s in figure 9 in ADCP coordinate transformation booklet – (3) added in line 102-103

(2) N-BoB site : 6.99 cm/s (see picture below) – (3) added in line 122-123

[Figure]

(1) m6 (Lines 136-137): "We then solved the Sturm-Liouviille equation (4) and... and amplitude." This sentence might read more naturally if it is after equation 4 is introduced.

(2) Yes.

(3) The sentence was removed.

(1) m7 (Line 426): "...Peru shelf, a location with a strong internal wave field". A reference for this statement would be helpful. I do not think Lentz and Chapman (2004) mention this.

(3) A reference to the Jackson 2004 Internal waves atlas that gives pictures and description of internal waves packets on the peruvian shelf was added. I also modified the sentence in line 425 into :

« Lentz and Chapman (2004) » presented observations made around the world, including negative volume flux near the seabed under upwelling on the Peru shelf — a location characterized by regular internal wave packets « Jackson 2004 ».

(1) m8: The axis labels and some of the annotations in most figures are too small to read without zooming in.

(3) The axis labels and annotations in Figure 1, 2, 3, 4 ,5 have been increased.

Typos/minor edits

Line 11: an -> a

ok

Line 38: transportation -> transport

ok

Line 89: over -> in

ok

Line 159: of the -> the

ok

Line 189: in the next -> in

ok

Line 197: this -> that

ok

Line 250 isotherms -> isotherm

ok

Line 281: maxima -> maximum

ok

Line 297: dominate -> dominant

ok

Line 301: waves -> wave

ok

Line 306: height -> heights

ok

Line 319: Qr1 -> Qr

ok

Line 321: T > 15°C -> T < 15°C; dominatly -> dominantly

Thank you

Line 352: drove Q_A asymmetry -> caused Q_A's asymmetry

ok

Lines 358, 387, 389, and elsewhere: NLIW -> NLIWs

ok

Line 361: time series -> the time series

ok

Line 387 attributed -> attributed to

ok

**Uncertainty on filtered transport**

September 13, 2025

**1  Context**

This section aim to answer the following comment regarding the paper submitted to EGU-2025-2072 :

M1: Since the transports Q, QEu and Qr (and other quantitative estimates) are based on measurements which contain random error, it is important to have error bounds derived from the propagation of the instrumental noise. This would add confidence to the results, for example the comparison between experimental and theoretical Stokes drift fluxes in Section 4.1. Figures 1(c,e), 6, 9, and 10 would also benefit from this analysis.

**2  Method**

The first two sections list the general definition used to compute the propagation of uncertainty measurements.

**2.1  General statistics**

We recall the definition of the variance. We use it to express the uncertainty due to filtering in section 2.6.

**2.1.1  Measured value vs. true value**

We assume that the true physical quantity is denoted by $X_{\text{true}}$. The instrument provides a noisy measurement

$$X_{\text{mes}} = X_{\text{true}} + \delta_X \tag{1}$$

where $\delta_X$ is the measurement error (assumed to have zero mean if the instrument is unbiased).

**2.1.2  Variance as the ensemble average of squared errors**

The variance of the measurement error is defined as

$$\sigma^2 = \mathbb{E}\big[\,(X_{\text{mes}} - X_{\text{true}})^2\,\big] = \mathbb{E}\big[\delta_X^2\big], \tag{2}$$

that is, the ensemble average of the squared differences between the measured value and the true value.

**2.2  The general propagation of uncertainty**

We use the propagation of errors formula to express the error on a quantity $F$ calculated from a combinaison of $n$ variables $x_i$.

$$\sigma_F^2 = \left(\frac{\partial F}{\partial x_1}\right)^2 \sigma_{x_1}^2 + \left(\frac{\partial F}{\partial x_2}\right)^2 \sigma_{x_2}^2 + ... + \left(\frac{\partial F}{\partial x_N}\right)^2 \sigma_{x_N}^2 \tag{3}$$

The propagation error formula can be used in case of small error terms and uncorrelated, independant variables. It can be found using equation 1, more information on the formula in section 3.16.8 in the book [Emery and Thomson 2001 - ISBN 978-0-12-387782-6].

**2.3 Total transport**

The total transport express as :

$$Q(t,T) = < \int_{z(T_1,t)}^{z(T_2,t)} u(z,t) \, dz > \tag{4}$$

Where $z(T_i, t)$ is the height for which $T(z,t) = Ti$ and $u(z,t)$ is the horizontal cross-shelf velocity. The temperature and the velocity are independant and coloacted on a regular grid with vertical spacing $\delta_h$. Notation $< . >$ represente low-pass filter.

**2.3.1 The uncertainty on unfiltered transport**

The unfiltered transport express as :

$$Q'(t,T) = \int_{z(T_1,t)}^{z(T_2,t)} u(z,t) \, dz \tag{5}$$

The discrete form is :

$$Q'(t; T_1, T_2) = \delta_h \sum_{k=k_1}^{k_2} u_k(t) \tag{6}$$

With $\quad M = k_2 - k_1 + 1$. and $k$
Using equation 3 on equation 6 we find :

$$\sigma_{Q'}^2(t) = \delta_h^2 \sum_{k=k_0}^{k_1} \sigma_u^2 + u(z_{(T_1)}, t)^2 \sigma_{z_{(T_1)}}^2 + u(z_{(T_2)}, t)^2 \sigma_{z_{(T_2)}}^2 \tag{7}$$

$$\sigma_{Q'}^2(t) = M \delta_h^2 \sigma_u^2 + u(z_{(T_1)})^2 \sigma_{z_{(T_1)}}^2 + u(z_{(T_2)})^2 \sigma_{z_{(T_2)}}^2 \tag{8}$$

with $\sigma_u$ is the standard deviation on one ping for a given ADCP configuration and $\sigma_z$ is the uncertainty on the temperature height. The variance $\sigma_{z(T_i)}$ is defined in the next section.

**2.4 Inteprolation and filter**

**2.4.1 The uncertainty on the temperature height**

The uncertainty on the temperature height is denoted $\sigma_{z(T_i)}$. It depends on the temperature and pressure measurement noise and on the temperature interpolation. The measurements from sensors $a$ and $b$ are denoted $(T_{ma}, z_{ma})$ and $(T_{mb}, z_{mb})$. Between two sensor the temperature is linearly interpolated. The inteprolated temperature field is denoted $Ti$ and express as :

$$T_i = (z - z_a) \frac{T_b - T_a}{z_b - z_a} + T_a \tag{9}$$

We define $\quad H = z_b - z_a, \ D = T_b - T_a, \ N = T_i - T_a$, so that :

$$z(T_i) = z_a + H \frac{N}{D} \tag{10}$$

The variance is computed using equation 3 and we note $\sigma_{T_a} = \sigma_{T_b} = \sigma_{T_m}$ and $\sigma_{z_a} = \sigma_{z_b} = \sigma_{z_m}$.

$$\sigma_{z(T_i)}^2 = \sigma_{T_m}^2 \left( \left( \frac{z_b - z_a}{(T_b - T_a)^2} (T_a - T_i) \right)^2 + \left( \frac{z_b - z_a}{(T_b - T_a)^2} (Ti - Tb) \right)^2 \right) + \sigma_{z_m}^2 \left( \left( \frac{T_i - T_a}{T_b - T_a} \right)^2 + \left( 1 - \frac{T_i - T_b}{T_b - Ta} \right)^2 \right) \tag{11}$$

and using the expression defined above we can write :

$$\sigma_{z(T_i)}^2 = \sigma_{T_m}^2 \frac{H^2}{D^4} ((N - D)^2 + N^2) + \sigma_{z_m}^2 \left( \left( 1 - \frac{N}{D} \right)^2 + \left( \frac{N}{D} \right)^2 \right) \tag{12}$$

The contribution of $\sigma_{T_m}$ increases when the distance between the two measurementspoints increases, and when $D$ decreases, i.e when the two measurements are similar. The contribution of $\sigma_{z_m}$ is bounded by a multiplication factor 0.5 when $T_i = \frac{T_b + T_a}{2}$ and 1, when $T_i = T_a$ or $T_i = T_b$. Therefore, the main source of error in the temperature height is expected to be the temperature measurement. This error is reduced when the sensors are close to each other but distant enough to measure distinct temperature.

**2.4.2 The uncertainty due to filtering**

The filter used is a Butterworth filter of order 3 from scipy.signal python package. We applied it on a time series with a sampling frequency $fs$ which depends on the data set (N-BoB or SE-BoB) and a cutting frequency $f_c$ of 24 hours. The filter impulse response is denoted $h(t)$ (see Figure 1) and is applied forward and backward using scipy.signal.filtfilt().

[Figure]

Figure 1: Caracteristics of the Butterworth low pass filter of 3rd order used on SE-BoB dataset : the cutting frequency is $f_c = 1/(24.60)$ and the frequency sampling is $fs = 1$.

We denote the unfiltered transport $Q'$ and $\delta_{Q'}$ the measurement error. The error is non stationnary.

$$Q(t) = h * Q'(t) = \int_{-\infty}^{\infty} h(\tau)Q'(t - \tau)d\tau \tag{13}$$

Using the measured value vs. true value concept explicited in section 2.1.1 it can be shown that :

$$\delta_Q(t) = \int_{-\infty}^{\infty} h(\tau)\delta_{Q'}(t - \tau)d\tau$$

$$\mathbb{E}[\delta_Q^2(t)] = \iint_{-\infty}^{\infty} h(\tau)h(\tau')\mathbb{E}[\delta_{Q'}(t - \tau)\delta_{Q'}(t - \tau')]d\tau d\tau'$$

Where $\mathbb{E}$ is the ensemble average. We assume the errors to be $\delta$-correlated so that :

$$\mathbb{E}[\delta Q(t - \tau)\delta Q(t - \tau')] = \sigma_{Q'}^2(t - \tau)\delta(\tau - \tau')$$

Where $\delta(t)$ is the Dirac function and therefore :

$$\sigma_Q^2 = \iint_{-\infty}^{\infty} h(\tau)h(\tau')\sigma_{Q'}^2(t-\tau)\delta(\tau-\tau')d\tau d\tau' \tag{14}$$

$$\sigma_Q^2 = \int_{-\infty}^{\infty} h^2(\tau)\sigma_{Q'}^2(t-\tau)d\tau \tag{15}$$

The variance $\sigma_Q^2$ is obtained by filtering the variance of the unfiltered transport $\sigma_{Q'}^2$ using the squared impulse response of the filter (see Figure 1).

**2.5   Eulerian transport**

Eulerian transport is computed over filtered velocity $< u(z,t) >$ and the filtered temperature field. The height of the filtered isotherm is denoted $z_{Eu}(T,t)$. We can write the Eulerian transport as :

$$Q_{Eu}(t,T) = \int_{z_{Eu}(T_1,t)}^{z_{Eu}(T_2,t)} < u(z,t) > dz \tag{16}$$

Using equation 3 we can write the uncertainty on Eulerian transport as :

$$\sigma_{Q_{Eu}}^2(t,T) = M\delta_h^2\sigma_{}{}^2 + ^2 \sigma_{z_{Eu(T_1)}}^2 + ^2 \sigma_{z_{Eu(T_2)}}^2 \tag{17}$$

where $M$ and $\delta h$ definition are given in section 2.3. With the uncertainty due to filtering expressed in section 2.4.2 we can write :

$$\sigma_{}^2 = \int_{-\infty}^{\infty} h^2(\tau)\sigma_u^2(t-\tau)d\tau \tag{18}$$

$$\sigma_{z_{Eu}}^2 = \int_{-\infty}^{\infty} h^2(\tau)\sigma_z^2(t-\tau)d\tau \tag{19}$$

**2.6   Residual transport**

Residual transport is the difference between Eulerian and total transport. We masked the residual transport for which the Eulerian and total transport were untrustable (error up to 50 % of the value).

**3   Results**

**3.1   Total transport**

We present the uncertainty on transport for the SE-BoB dataset. Equivalent work could be done on the N-BoB dataset. At the SE-BoB $\sigma_u = 0.08m/s$, $\sigma_{Tm} = 0.1C$ and $\sigma_{zm} = 0.1m$.

**3.1.1   Error contribution before filtering**

We identify which term contribute the most to the uncertainty in equation 8, between $\sigma_u$ and $\sigma_{z(T_i)}$ (Figure 2). The velocity error dominates near the boundary, where the temperature is the measured temperature. In the interior, the Temperature/Pressure error dominates most of the time due to the distance between the sensors ($H$=10 m in SE-BoB dataset). At high frequency the velocity error can dominates (Figure 2). The velocity error dominates regularly at the same temperature range. This is likely due to an isotherm that regularly matched one of the temperature sensor and likely close to a stratified location. The values are set to NaN when no transport is measured or it is smaller than the error. To apply the filter we filled the variance time series with it's time average.

The maximum variance of the total transport after the 24h low pass filter is 0.035 $m/s$ (Figure3). The maximum values are between 16 and 20 $C$ before the 17/07, when the transport reach a maximum of $-1.4$ $m/s$ (Figure 4).

[Figure]

Figure 2: Error type at each time step and each temperature range from equation 8.

[Figure]

Figure 3: The variance on total transport $\sigma_Q$ due to the propagation of the measurement uncertainties

**3.1.2 Total transport time series and profile**

Where the variance reached more than 50% of the transport we masked the transport (Figure4). The main structures were not masked.

**3.2 Eulerian transport variance and mask**

The variance was smaller for Eulerian transport than on total transport. The maximum variance on Eulerian transport was $0.008m/s$ (Figure 5). For ease of computation $\sigma_{z_{Eu}}$ was computed using a constant $\sigma_z = 0.7m$, obtained from equation 12 with $H = 10m$, $D = 1C$ and $N = 0.5C$. These values were chosen from the average temperature profile and sensor position shown in Figure 1 of the manuscript. Due to filter, uncertainty was reduced $\sigma_{z_{Eu}} = 0.02m$.

We masked the transport where the variance reached more than 50% of the transport (Figure6).

[Figure]

Figure 4: (Top) Total transport masked when the error is more than 50 % of the value (bottom) Original transport

[Figure]

Figure 5: Variance on Eulerian transport as a function of time and temperature range. For the SE-BoB site.

Almost no data were masked on the Eulerian field.

**3.2.1 Residual transport time series and vertical profile**

The variance in residual transport was dominated by the variance on total transport. Based on Figure 3 the variance in residual transport was of the same order : $10^{-2}m/S$.

We applied the two masks from the total transport and Eulerian transport analyses to residual transport (Figure 7). Positive transport near the bottom (where the temperature is below 14°C) was partly masked.

The time average was mostly affected by the mask mostly for temperatures below 14°C (Figure 8 9). Where the transport was nan we considered the tranpsport to be nill. The positive transport for low temperature remained positive but was reduced between the original and masked transport.

[Figure]

Figure 6: (Top) Eulerian transport masked when the error is more than 50 % of the value (bottom) Original eulerian transport

[Figure]

Figure 7: (Top) Residual transport masked with the combined mask from total and eulerian analysis. (bottom) Original residual transport

**4    Conclusion and acknowledgement**

Most of the variance in the measurements was due to temperature. The main source of uncertainty in residual transport was the uncertainty in total transport. Masking the value did not remove the main structures observed and described in the paper. The vertical profile of residual transport remained unchanged. The analysis confirmed the results observed in the manuscript.

We would like to express our sincere gratitude to Louis Marié (Ifremer LOPS) for his invaluable assistance in addressing the impact of filters on uncertainty measurements. We also acknowledge the use of the GPT-4o model in developing the code.

[Figure]

Figure 8: Vertical profile of original residual transport (black) and masked residual transport (red) averaged on the total time series

[Figure]

Figure 9: Vertical profile of original residual transport (black) and masked residual transport (red) averaged on the 1st spring tide

[Figure]

Figure 10: Vertical profile of original residual transport (black) and masked residual transport (red) averaged on the 1st spring tide

---

## Author Response (AR2)

**answer to editor comments**

October 2025

Dear Authors Thank-you for your responses to referees and the revised manuscript. I think you have largely responded appropriately to the referee comments but in some cases I would like to see better representation of your response in the revised manuscript. If readers have a similar comment it is better to give them easy access to some form of response, either directly in the manuscript or by reference. This applies to the first three items below. The other "Detailed comments" are minor but are intended to improve clarity in a revised version which I would like to see.

Yours sincerely

John Huthnance (Editor)

Dear Editor,

We thank you for your interest in our paper and your valuable comments.

We are pleased to present our revised manuscript for your consideration. Among the main changes, we included, as suggested, some elements of our response to the reviewers directly in the manuscript.

A detailed point-by-point response to your comments is appended to this letter.

Best regards, Adèle Moncuquet

**1 Editors comments**

Regarding Ref 1 comment on Fig. 1 and comparison between mooring and MVP data, I think you should consider including a summary of your response in the final manuscript.

We have added this paragraph in the revised manuscript (line 101-104) :

"In a previous paper using the same dataset (Moncuquet et al., 2025), the vertical distribution of the sensors was shown to be satisfactory to capture the stratification. A comparison of the temperature profiles from the thermistors and the higher-resolution MVP measurements is presented in Figure 7 of that study."

Regarding Ref 2 comment M1, I think you should consider including your response "M1.pdf" as a linked supplement and also include its results briefly

in the main text with reference to the supplement. Please note however that supplements are not copy-edited.

We have added a paragraph in both the Methods section and the Conclusions section of the revised manuscript. We have also included M1.pdf as Supplementary material.

We added this text (line 200-204): "We computed an estimate of the uncertainty in both the total and the Eulerian transport by propagating the measurement uncertainties (see Supplementary material for details). The uncertainty was on the order of $10^{-2}m^2/s$. The uncertainty was generally small compared to the transport magnitude and only a small number of near-zero values had an uncertainty that was greater than 50% of the transport. However, removing these values resulted in time-averages that were slightly reduced, therefore we decided to not remove these values from the analysis to avoid introducing bias."

We added this text (line 452-455) : "Instrument uncertainty propagation analysis showed that uncertainty was generally small compared to the transport magnitude ($10^{-2}m^2/s$) and only a small number of near zero values had an uncertainty that was greater than 50% of the transport. This demonstrates the robustness of both the residual transport structure and magnitude (see Supplementary materials)."

Regarding Ref 2 comment M3 about the 24 h filter cutoff, I think you should consider including a summary of your response in the final manuscript.

We added these sentences to the Methods section (line 195-198): "The 24-hour low-pass filter was evaluated against the Demerliac tidal filter (Demerliac,1974), applied to hourly data, and was found to perform adequately (see Appendix Figures A3,A4,A5).The 24-hour low-pass filter was selected as it can be applied to non hourly data and maintain the full temporal coverage of the measurements."

We have added the 3 figures in the response to M3 and this paragraph in the Appendix : "Eulerian transport was computed using a 24-hour low-pass filter and Demerliac tidal filtering applied to hourly data. The time-averaged $Q_{Eu}$ was similar with both methods. Although the filtered Demerliac data display less variability, the structure and intensity of the observed events are similar to those in the 24-hour low-pass dataset, which was considered adequate for the current analysis."

**2    Detailed comments**

- Line 38 end. "boluses"?
    Corrected
- Line 102. "so that the horizontal" needs completion or deletion.
    Corrected

- Line 108. Better "between 48.5°N and around 46.5°N" or "from 48.5°N to around 46.5°N". Similarly for line 251 dates.

Corrected

- Lines 130-131. I think you rotated the co-ordinates, not the currents.

Corrected

- Line 162. "an" implies just one "NLIW" (omit "an" if more than one) whereas "NLIWs" implies more than one.

Corrected here and elsewhere, - Line 200. Does "that paper" refer to Franks et. al (2020) or Thorpe (1968)? Better to be explicit.

Corrected

- Line 230. "offshore" – "onshore"?

Yes, thank you so much.

- Line 231. "August 10" is long before the start of figure 2b.

Yes I meant October 8. I'm so sorry thank you.

- Line 241 end. Better "between 20 and 15 mab." or "at 20-15 mab."

Corrected

- Line 251. C.f. line 108 comment.

Corrected

- Lines 268-269. I suggest "flow; Lentz" and "2004) from July 12- Figure 4a,b."

Changed

- Lines 271-272. Just before July 27 as well? "increase of the temperature . . . warming of the near seabed temperature" – duplication.

It started just before July 27, I modified the text and removed the parenthesis to suppress the duplication.

- Line 303: "internal waves solitons, surfaces wave," –¿ "internal wave solitons, surface waves,"

Corrected

- Line 311. "on" – "to"

Corrected

- Line 315. Better ". . 15 °C; QEu was small . ."

Changed

- Line 353: "drove" - "caused"

Changed

- Line 377. ". . (Figure 7a,b). . ."

Changed

- Line 467. ". . NLIWs on the different components . ."?

Corrected

- Line 472. ". . tides and NLIWs . ."?

Corrected

- Lines 491-493. "using" three times in the sentence. It is not clear what two variables the regression is between. Agreed. The sentence was modified into : "Density was computed as a function of the temperature using the linear regression observed from the MVP measurements. The relationship between density and conductivity was considered stationary. "

**3   M1.pdf**

- 1 Context. "This supplement aims . . comment: "Since . . fluxes." would suffice for the final publication.

   ok

- 2.2 first sentence. ". . error of a quantity . . combination . ."

   Corrected

- Second line after (3). ". . formula is in . ."

   Corrected

- 2.3 before (4) and 2.3.1 before (5). ". . transport is expressed as . ."

   Corrected

- Second line after (4). ". . independent and collocated . ."

   Corrected

- Third line after (4). ". . represents low-pass filtering."

   Corrected

- Line after (6) I think you want the definition of $\delta h$ here.

   yes, we have added the definition of $\delta h$; $k$ and moved the definition of $M$ at this location.

- Line before 2.4 and line of 2.4.1. "uncertainty in"

   Corrected

- Line before (9). "interpolated" (spelling). "expressed as".

   Corrected

- 2.4.2 line 1. Better to mention "low pass" here as well as in the figure 1 caption.

   Changed

- Line 3 and Figure 1 caption. "cutting" –¿ "cut-off"? Line 3 also "1/(24 hours)".

   Corrected

- Figure 1 should give units for frequency.

   Corrected

- Line after (13). "explicited" –¿ "described"

   Corrected

- 2.6 second sentence. "untrustable" – "uncertain"?

   Corrected

- 3.1.1 lines 4-5. "can dominate".

   Corrected

- 3.1.1 2nd paragraph 2nd sentence. ". . before 17th July, when the transport reached a maximum"

   Corrected

- 3.2 line 5: ". . reduced: $\sigma$ . ."

   Corrected

- 3.2.1 line 6. Omit first "mostly". Line 7. "NaN"

   Changed